# mTORC1 hampers Hedgehog signaling in *Tsc2* deficient cells

Lasse Jonsgaard Larsen[1], Elsebet Østergaard[1,2], Lisbeth Birk Møller[1]

The mTORC1-complex is negatively regulated by TSC1 and TSC2. Activation of Hedgehog signaling is strictly dependent on communication between Smoothened and the Hedgehog-signaling effector and transcription factor, GLI2, in the primary cilium. Details about this communication are not known, and we wanted to explore this further. Here we report that in *Tsc2*$^{−/−}$ MEFs constitutively activated mTORC1 led to mislocalization of Smoothened to the plasma membrane, combined with increased concentration of GLI2 in the cilia and reduced Hedgehog signaling, measured by reduced expression of the Hedgehog target gene, *Gli1*. Inhibition of mTORC1 rescued the cellular localization of Smoothened to the cilia, reduced the cilia concentration of GLI2, and restored Hedgehog signaling. Our results reveal evidence for a two-step activation process of GLI2. The first step includes GLI2 stabilization and cilium localization, whereas the second step includes communication with cilia-localized Smoothened. We found that mTORC1 inhibits the second step. This is the first demonstration that mTORC1 is involved in the regulation of Hedgehog signaling.

## Introduction

The mammalian target of rapamycin complex 1 (mTORC1) is part of the PI3K/Akt/mTOR pathway, controlling cell growth, metabolism, and autophagy in response to growth factors, nutrients, and energy levels (1, 2). Tuberous sclerosis complex (TSC, OMIM: #191090), a severe autosomal dominant disorder, is caused by disease-causing variants in either of the tumor suppressor genes, *TSC1* (OMIM: #605284) or *TSC2* (OMIM: #191092). The gene products of *TSC1* and *TSC2*, named hamartin/TSC1 and tuberin/TSC2, respectively, form a protein complex that, in the absence of sufficient amounts of nutrients, inhibits mTORC1 via inhibition of the mTORC1 activator RHEB (3). Inhibition of mTORC1 leads to the activation of autophagy (4).

TSC is characterized by constitutive mTORC1 activation and the development of benign tumors (hamartomas) in many organs among many other symptoms (5). Increased mTORC1 activity has also been linked to carcinogenesis and tumorigenesis (6).

In the presence of sufficient amounts of nutrients, mTORC1 is activated as a result of AKT-mediated phosphorylation of TSC2, leading to inhibition of the TSC1/TSC2 complex. Activated mTORC1 phosphorylates several substrates including the ribosomal S6 kinase 1 (S6K1) and the eukaryotic translation initiation factor 4E (eIF4E)-binding protein 1 (4E-BP1) (7, 8). Phosphorylation of 4E-BP1 releases its binding from eIF4E, enabling incorporation of eIF4E into the eukaryotic initiation factor 4F (eIF4F) protein complex to initiate cap-dependent translation. The 40S ribosome protein S6 (S6) is a direct substrate of S6K, and the presence of phosphorylated S6 (pS6) is a reliable indicator of mTORC1 activity (9).

We have previously demonstrated crosstalk between mTORC1 and Hedgehog (Hh)-signaling pathways (10), but besides this not much is known about the interaction between the two signaling pathways (11). The primary cilium, an antenna-like structure which extends from the cell surface of almost all quiescent cells, coordinates a large number of signaling pathways, including mTORC1 and Hh signaling (10, 12). The centrosome contains the mother and daughter centrioles, and it coordinates both the growth of the primary cilium and spindle pole formation during mitosis. The presence of the primary cilium is therefore incompatible with cell division. In quiescent cells, the mother and daughter centrioles migrate to the plasma membrane to form the basal body, enabling the growth of the primary cilium. When the cell enters the cell cycle, the primary cilium is disassembled and the centrioles separate to coordinate the assembly of the bipolar spindles (13).

Canonical Hh signaling is very complex: In the absence of Hh-ligand, Patched 1 (PTCH1), the receptor of Hh, is active inside the primary cilia, posing a constitutive inhibition on SMO. When Hh-ligand binds to PTCH1, PTCH1 activity is blocked, allowing SMO activation and accumulation in the primary cilia. Moreover, transcriptional activation of target genes in Hh signaling is restricted because of ubiquitination-mediated degradation of the transcription factors GLI1, GLI2, and GLI3 (14). In the presence of Hh-ligand, PTCH1 is displaced away from the primary cilium, allowing

[1]Department of Genetics, Kennedy Center, Copenhagen University Hospital, Rigshospitalet, Glostrup, Denmark   [2]Department of Clinical Medicine, University of Copenhagen, Copenhagen, Denmark

Correspondence: Lisbeth.Birk.Moeller@Regionh.dk

the accumulation of Smoothened (SMO) in the ciliary membrane (15), and stabilization of the full-length (FL) GLI2 and GLI3 proteins (GLI2-FL and GLI3-FL), which subsequently leads to translocation of the activator forms (GLI2-A and GLI3-A) to the nucleus (16). Similar scenarios of cilia localization of SMO occur when using SMO agonists such as purmorphamine for Hh activation (17).

GLI2-A is the transcription factor mainly responsible for the transcriptional activation of Hh-target genes, which include *GLI1*. Ciliary localization of both SMO and GLI2 is required for GLI2-dependent Hh-induced target gene transcription (11, 18), but details about GLI2 activation are not known. The Hh-signaling pathway is fundamental for embryonic development and for postnatal tissue homeostasis, renewal, and regeneration (19, 20, 21). Dysregulated Hh-pathway activity gives rise to birth defects, including left–right asymmetry of vertebrate embryos. Furthermore, dysregulated Hh has, in similarity to mTORC1, been linked to carcinogenesis, and several Hh-signaling pathway inhibitors have been developed for cancer treatment (22).

Previously, we demonstrated a reduced Hh-induced expression of *Gli1* mRNA in murine cells without functional *Tsc1* ($Tsc1^{-/-}$) or *Tsc2* ($Tsc2^{-/-}$) (10). We showed that the reduced Hh signaling in *Tsc1*-null cells is caused by low expression of *GLI2*, which is regulated by SMAD2/3, because TSC1 is required for phosphorylation and thereby activation of SMAD2/3. Whereas the TSC1-dependent effect on Hh is known, not much is known about the TSC2-dependent effect on Hh signaling (10).

Here, we have investigated the effect of *Tsc2* on Hh signaling in detail and reveal a previously unknown interaction between mTORC1 and Hh signaling. Our data indicate that reduced Hh activation in *Tsc2* null cells is because of hampered GLI2 activation and nuclear translocation and that this might be a result of impaired ciliary location of SMO. Our results support previous studies demonstrating that the activation and nuclear translocation of GLI2 is dependent on cilia-located activated SMO but, furthermore, demonstrate a hitherto unknown impact of mTORC1 on SMO trafficking and GLI2 activation. Our results reveal evidence for a two-step activation process of GLI2.

Together, our present and previous studies reveal major differences in crosstalk between TSC and Hedgehog signaling, and the effect of mTOR inhibitors, depending on whether *Tsc1* or *Tsc2* is absent. Based on these results, we suggest including the genetic background in future studies of the treatment of patients with TSC because it may explain possible differences in treatment response.

# Results

### *Tsc2* deficiency impairs Hh signaling

We have previously demonstrated that canonical Hh signaling, which is strictly dependent upon functional primary cilia, is impaired in $Tsc2^{-/-}$ MEFs (10). To confirm this, $Tsc2^{-/-}$ MEFs were cultured for 48 h in a serum-reduced medium to induce growth arrest and cilia formation, and to activate Hh signaling, the cells were stimulated with the SMO agonist purmorphamine (Pur). In agreement with our previous studies (10), we observed a

significantly reduced Hh-response in the $Tsc2^{-/-}$ cells compared with wild-type (WT) cells, determined by the expression level of the Hh-induced target gene *Gli1* (Fig 1A). Because Pur bypasses PTCH1 by targeting SMO directly, we wanted to evaluate the effect of PTCH1 by Hh activation of the cells using Sonic Hedgehog (SHh) conditioned medium. As seen in Fig 1B, also when stimulating the cells with SHh, we observed a significantly lower level of *Gli1* expression in the $Tsc2^{-/-}$ cells compared with the WT cells, indicating that the inclusion of PTCH1 did not resolve the defect. Although the expression level of *Gli1* was lower in the $Tsc2^{-/-}$ cells compared with the WT cells, the SHh-induced increased expression of *Gli1* mRNA in the $Tsc2^{-/-}$ cells, verifying that the cells were Hh-signal responsive. Indeed, as a result of the significantly lower basal level of *Gli1* mRNA in the $Tsc2^{-/-}$ cells, compared with the WT cells, the fold increase in expression of *Gli1* mRNA after Pur and SHh stimulation was significantly higher in the $Tsc2^{-/-}$ cells compared with the WT cells (Fig S1). To confirm that the lower Hh-induced expression level of *Gli1* mRNA in the $Tsc2^{-/-}$ MEFs was an effect of lack of functional *Tsc2*, we transfected $Tsc2^{-/-}$ MEFs with the plasmid pTSC2, encoding the human TSC2 protein. Using this strategy, we found that the exogenously expressed *TSC2* indeed was able to rescue Hh signaling (Fig 1C). As an alternative approach, we also studied the effect of *Tsc2* on Hh signaling by down-regulation of *Tsc2* in WT cells. After transfection of WT MEFs with siRNA against *Tsc2*, we observed a significant reduction in the Pur-induced expression level of *Gli1* mRNA as an effect of the *Tsc2* siRNA treatment (Fig 1D). No reduction in the basal level of *Gli1* mRNA was observed, possibly because *Tsc2* siRNA treatment does not lead to a total loss of TSC2 (Fig 1D, insert).

Taken together, these results demonstrate that Hh signaling was negatively influenced by *Tsc2* deficiency in the MEF cell line and that PTCH1 was not involved, as a similarly reduced *Gli1* expression was observed with both Pur and PTCH1 dependent SHh stimulation.

### Rapamycin treatment rescues canonical Hh signaling in $Tsc2^{-/-}$ MEFs

Ciliogenesis is induced by autophagy (12, 23), and inhibition of mTORC1 initiates the autophagic process (4). Because TSC2 is a well-known negative regulator of mTORC1, the lack of a functional *TSC2* gene is characterized by constitutive mTORC1 activation even at low nutrient levels, compromising autophagy (24). We have previously demonstrated that the impaired Hh signaling in $Tsc2^{-/-}$ MEFs is restored by the mTORC1 inhibitor rapamycin (Rapa). To confirm this, we treated $Tsc2^{-/-}$ MEFs with a combination of Pur and the mTORC1 inhibitor Rapa. As shown in Fig 2A, in agreement with our previous results (10), Rapa rescued Hh signaling, indicating that the *Tsc2*-dependent impairment of Hh signaling is a result of increased mTORC1 activity.

To test if the same picture was obtained by targeting PTCH1, we treated $Tsc2^{-/-}$ MEFs with a combination of SHh-conditioned medium and Rapa. Using this approach, Rapa also rescued Hh signaling, verifying that mTORC1 affects Hh-signaling downstream of PTCH1 (Fig 2B).

To verify that Rapa leads to inactivation of mTORC1, the level of the mTORC1 indicator, pS6, in $Tsc2^{-/-}$ MEFs was analyzed by immunofluorescence microscopy (IFM). The absence of pS6 in the

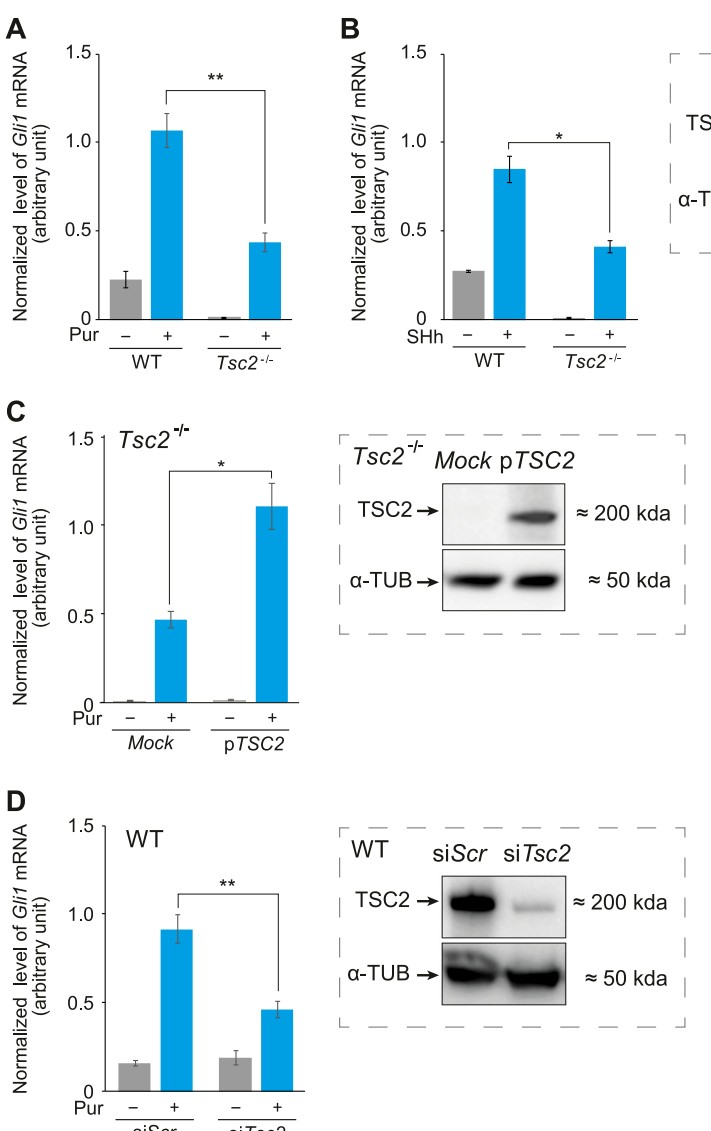

**Figure 1. Tsc2 deficiency impairs the activation of Hh signaling.**
**(A)** Significantly reduced Hh signaling, measured by the expression level of *Gli1* mRNA, was observed in *Tsc2*⁻/⁻ MEFs compared with WT after stimulation with the Smo agonist, purmorphamine (Pur) (n = 4). **(B)** Significantly reduced Hh signaling was observed in *Tsc2*⁻/⁻ MEFs compared with WT, after stimulation with Sonic Hh-conditioned media (SHh) (n = 3). **Insert**: TSC2 is absent in *Tsc2*⁻/⁻ MEFs. Cell lysate from *Tsc2*⁻/⁻ and WT MEFs was analyzed by Western blotting (WB), using antibodies against TSC2 and α-tubulin (α-TUB), as indicated. **(C)** Exogenous expression of TSC2 was able to complement the compromised Hh signaling in *Tsc2*⁻/⁻ MEFs. *Tsc2*⁻/⁻ MEFs were transfected with *TSC2* (pTSC2) or Mock-expressing plasmid, followed by Hh stimulation (Pur) (n = 2). **Insert**: Verification of exogenous expression of *TSC2* was performed by WB, using antibodies against TSC2 and α-TUB. **(D)** Down-regulation of *Tsc2* led to reduced Hh signaling. WT MEFs were transfected with siRNA targeting either *Tsc2* or a scramble sequence (Scr) before Hh stimulation (Pur) (n = 3). **Insert**: Verification of down-regulation of *Tsc2* was performed by WB, using antibodies against TSC2 and α-TUB. Furthermore, RT-qPCR demonstrated that the expression level of *Tsc2* was reduced to ~10% of the expression level in the control Scr cells (Fig S2). **(A, B, C, D)** The expression profiles of the target gene *Gli1* were normalized to *Tbp* mRNA expression (arbitrary units). Error bars represent SEM. *T* test, with significance levels *P < 0.05, **P < 0.01, and ***P < 0.001, was used. For specific *P*-values see Table S1.

*Tsc2*⁻/⁻ MEFs, upon Rapa treatment, confirmed that Rapa inhibits mTORC1 (Fig 2C). Note also that, when the cells were grown in complete medium, mTORC1 was active in both WT and *Tsc2*⁻/⁻, whereas when the cells were grown under serum-reduced conditions, mTORC1 was only active in *Tsc2*⁻/⁻ MEFs, confirming constitutive mTORC1 activation in *Tsc2*⁻/⁻ MEFs, as expected.

We then asked if the increased *Gli1* mRNA level in *Tsc2*⁻/⁻ cells treated with Rapa was a result of SMO-dependent canonical Hh signaling. To this end, the cells were treated with the SMO antagonist cyclopamine in combination with SHh and Rapa. As shown, cyclopamine completely abolished the induction of *Gli1* mRNA in Rapa- and SHh-treated cells (Fig 2D). Here, the Hh signaling was activated using an SHh-conditioned medium and not Pur because Pur and cyclopamine are competitive toward SMO binding (25). To further support this finding, *Smo* expression was down-regulated using siRNA, followed by treatment with Pur and Rapa. Down-regulation of *Smo* almost completely abolished the induction of *Gli1* mRNA (Fig 2E). Based on these results, we conclude that the

Rapa-mediated increase in *Gli1* expression in Hh-stimulated *Tsc2*⁻/⁻ MEFS is dependent on SMO.

As it is generally accepted that active mTORC1 suppresses autophagy and that basal autophagy regulates ciliary growth (26), we wanted to test if the increased *Gli1* expression, as an effect of Rapa treatment, was because of increased ciliation. We therefore calculated the fraction of ciliated *Tsc2*⁻/⁻ Pur-treated MEFs in the presence and absence of Rapa (Fig 2F). The ciliation was indeed increased in the Rapa-treated *Tsc2*⁻/⁻ MEFs. However, as the increase in ciliation was only ~17% (Fig 2F), and the *Gli1* mRNA level increased by ~70% (Fig 2B), we suggest that the increased ciliation cannot alone explain the increased Pur-mediated *Gli1* expression upon Rapa treatment. To test if the induction of *Gli1* mRNA, as an effect of mTORC1 inhibition, was a result of increased autophagy per se, we treated the cells with Trehalose, which triggers autophagy independently of mTORC1 (27), before Pur stimulation. Trehalose treatment did not increase the expression of *Gli1 mRNA* level in the *Tsc2*⁻/⁻ MEFs (Fig 2G), supporting that the increased *Gli1* expression

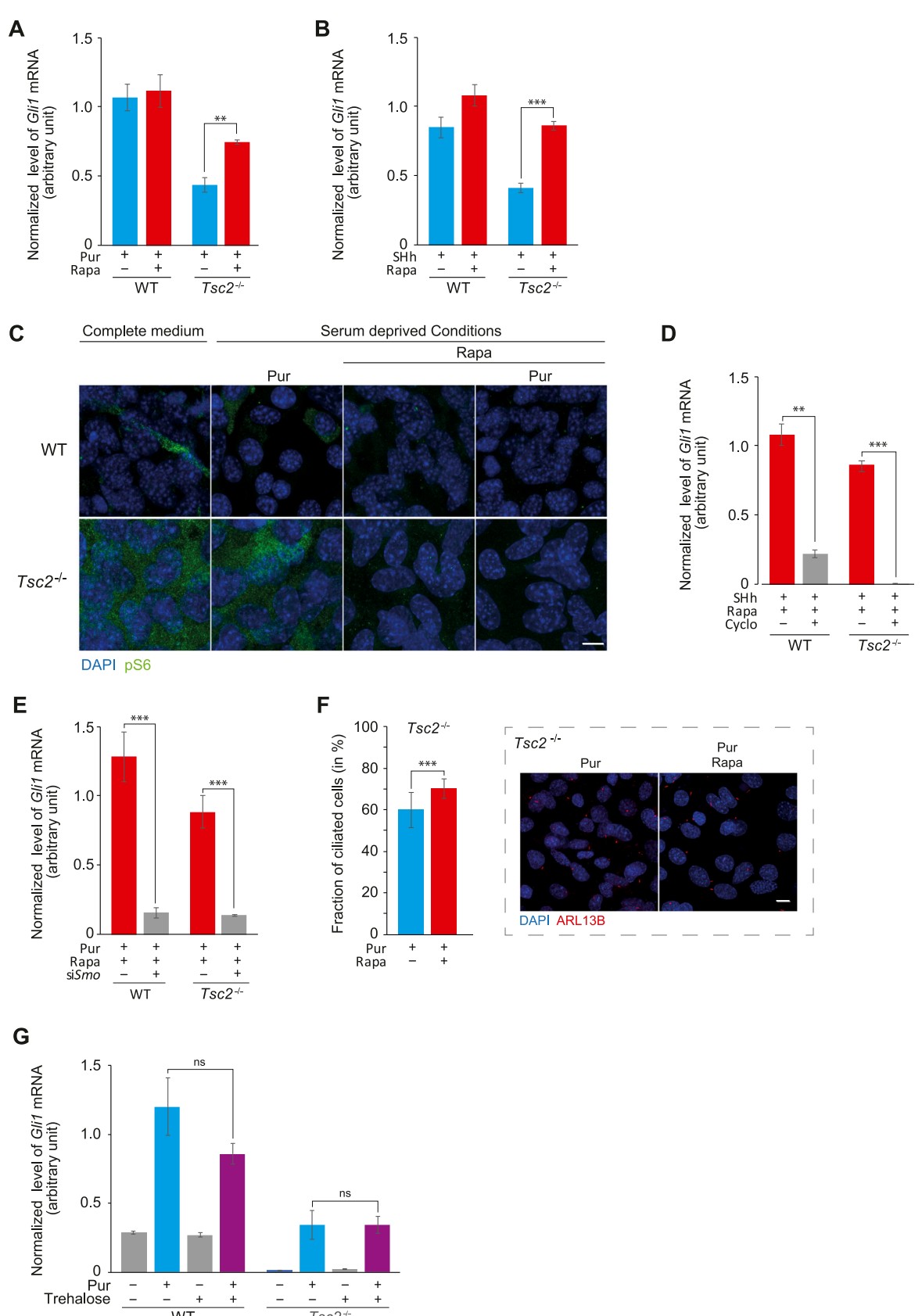

observed in the Rapa-treated cells (Fig 2A and B) was not a result of induced autophagy per se.

### Inhibition of mTORC1, but not mTORC2, restores canonical Hh signaling in $Tsc2^{-/-}$ MEFS

To verify that inhibition of mTORC1 leads to increased Hh signaling in $Tsc2$-deficient cells, we examined the effect of the mTOR inhibitor Torin1, which acts in a manner distinct from Rapa, being a selective ATP-competitive inhibitor of mTOR that inhibits both mTORC1 and mTORC2. In similarity to Rapa, treatment of $Tsc2^{-/-}$ MEFs with Torin1 led to an increase in $Gli1$ mRNA expression in response to Pur, demonstrating that inhibition of mTOR was essential for the induction of $Gli1$ mRNA (Fig 3A).

To further support an mTOR-dependent effect on Hh signaling, we treated the $Tsc2^{-/-}$ MEFs with siRNA against $mTOR$ before Pur stimulation. Consistent with an mTOR-dependent effect, downregulation of $mTOR$ significantly increased the Pur-mediated induction of $Gli1$ expression compared with $Tsc2^{-/-}$ MEFs treated with a scramble siRNA sequence ($Scr$) (Fig 3B).

As the primary target of Rapa is mTORC1, our results indicate that the mTOR-dependent induced $Gli1$ mRNA expression is mTORC1 specific. However, because it has previously been shown that long-term exposure to Rapa can affect the disassembly of mTORC2 (28) and Torin1 inhibits both mTORC1 and mTORC2, we cannot rule out an mTORC2-specific effect. Therefore, to distinguish between mTORC1- and mTORC2-specific effects, we treated the $Tsc2^{-/-}$ MEFs with siRNA against the mTORC1 component RPTOR and the mTORC2 component RICTOR, respectively. As only knockdown of $Rptor$ could rescue Hh signaling, the results indicate that the effect of $Tsc2$ deficiency on Hh signaling is mediated through mTORC1 and not mTORC2 (Fig 3C). In summary, these results verify that the impaired Hh signaling in $Tsc2^{-/-}$ MEFs in fact is mTORC1 dependent.

### Effect on downstream transcription factors

As the level of $Gli1$ mRNA is regulated by the transcription factor GLI2, we wanted to test if the increased $Gli1$ mRNA expression observed as an effect of Pur in combination with Rapa could be a result of increased amount of GLI2 protein. Investigation of the expression level of $Gli2$ mRNA in $Tsc2^{-/-}$ MEFs revealed that it was in fact increased as an effect of Rapa treatment, whereas no significant effect on the $Gli2$ mRNA level was observed as an effect of Pur

treatment. No effect on $Gli2$ mRNA expression of Pur or Rapa was observed in the WT MEFs (Fig 4A).

Moreover, because Hh activation is also dependent on SMO, we also investigated the effect of Pur and Rapa on the expression level of $Smo$ mRNA. No significant effect on $Smo$ expression was observed in $Tsc2^{-/-}$ or WT cells (Fig 4B).

To investigate if the increased level of $Gli2$ mRNA was accompanied by an increased amount of accumulated GLI2 protein, we investigated the cell lysate by Western blot analysis. Rapa treatment did not lead to any increase in GLI2 protein, indicating that the Rapa-induced increased expression of $Gli2$ mRNA did not lead to an increase in the accumulation of GLI2 protein (Fig 4C). However, in agreement with the previously demonstrated stabilization of GLI2 protein as a result of Hh activation (17), we observed substantially increased accumulation of GLI2 upon Pur treatment in both WT and $Tsc2^{-/-}$ MEFs.

Notably, the increased amount of GLI2 protein in the Pur-alone–treated cells did not affect the level of $Gli1$ mRNA, indicating that GLI2 is inactive, and that activation of GLI2 is dependent on inactivation of mTORC1 verified by increased $Gli1$ mRNA (revisit Fig 3A). GLI2 has been demonstrated to be degraded in response to PKA-promoted ubiquitylation, leading to reduced transcriptional activity (29). As we did not observe any significant effect of Rapa on the amount of accumulated GLI2 protein, it is unlikely that the observed reduced Hh-signaling output in $Tsc2^{-/-}$ cells was a result of increased GLI2 degradation. In agreement with this, inhibition of PKA by H89 did not lead to any increase in transcriptional Hh-output ($Gli1$ mRNA) (Fig 4D, left). Stimulation of PKA by the PKA activator forskolin led, as expected, to a decrease in Hh-output (Fig 4D, right).

### Aberrant cellular localization of GLI2 and SMO in $Tsc2^{-/-}$ MEFs

It has previously been demonstrated that activation of GLI2 is dependent on cilia membrane interaction and that the ciliary trafficking of GLI2 is triggered by the accumulation of activated SMO in the cilium (18, 30, 31, 32). Therefore, we proceeded to investigate whether the ciliary localization of GLI2 or SMO was affected in $Tsc2^{-/-}$ cells and if Rapa affected the localization.

As expected, we found a small number of GLI2-containing cilia, about 6% in the WT cells, in the absence of Hh activation (Fig 5A). Upon Hh activation by Pur, the number of GLI2-positive cilia in the WT MEFs increased to about 39% ($P$ = 2.2 × 10$^{-16}$). If the number of

**Figure 2. Hyperactivation of mTORC1 because of lack of $Tsc2$ impairs canonical Hh signaling.**
**(A)** Rapa rescued Pur-induced Hh signaling in $Tsc2^{-/-}$ MEFs. Pur stimulation was performed in the absence or presence of Rapa (n = 4). **(B)** Rapa rescued SHh-induced Hh signaling in $Tsc2^{-/-}$ MEFs. SHh stimulation was performed in the absence and presence of Rapa (n = 3). **(C)** $Tsc2^{-/-}$ MEFs are characterized by constitutively activated mTORC1, which is inactivated by Rapa. In WT MEFs, mTORC1 was inactivated also by serum deprivation. $Tsc2^{-/-}$ MEFs and WT cells were grown in complete medium or under serum-deprived conditions, respectively, in combination with Pur and Rapa, as indicated. Expression of phosphorylated S6 Ribosomal protein (pS6) was visualized by immunofluorescence microscopy using anti-pS6 antibody (green). The nuclei were visualized by DAPI staining (blue). Scale bar = 10 $\mu$m. **(D)** Smo antagonist hampered Rapa-mediated Hh-rescue. SHh-induced Hh signaling was investigated in the presence or absence of Cyclopamine (Cyclo) (n = 3). **(E)** Smo is required for Rapa-mediated Hh-rescue. Pur-induced Hh signaling was investigated after transfection with siRNA targeting $Smo$ (+) or as a negative control after transfection with $siScr$ (−) (n = 1). Verification of down-regulation of $Smo$ was performed by RT-qPCR analysis (see Fig S3, insert). **(F)** Rapa treatment leads to a modest increase in ciliation. Quantification of the fraction of ciliated cells, stimulated with Pur, was calculated in the presence or absence of Rapa. More than 1,000 cells, from three different experiments, were used for calculation. **Insert**: immunofluorescence microscopy analysis of the ciliation of $Tsc2^{-/-}$ MEFs, labeled with anti-ARL13B antibody (red). The nuclei were visualized by DAPI staining (blue). Scale bar = 10 $\mu$m. **(G)** The autophagy-inducer Trehalose did not rescue Pur-induced Hh signaling in $Tsc2^{-/-}$ MEFs. **(A, B, D, E, G)** The expression profiles of the target gene $Gli1$ were normalized to $Tbp$ mRNA expression (arbitrary units). Error bars represent SEM. $T$ test, significance levels *$P$ < 0.05, **$P$ < 0.01, and ***$P$ < 0.001, was used. **(F)** Error bars represent SEM. Fisher's exact test, with significance level *$P$ < 0.05, **$P$ < 0.01, and ***$P$ < 0.001, was used. For specific $P$-values, see Table S1.

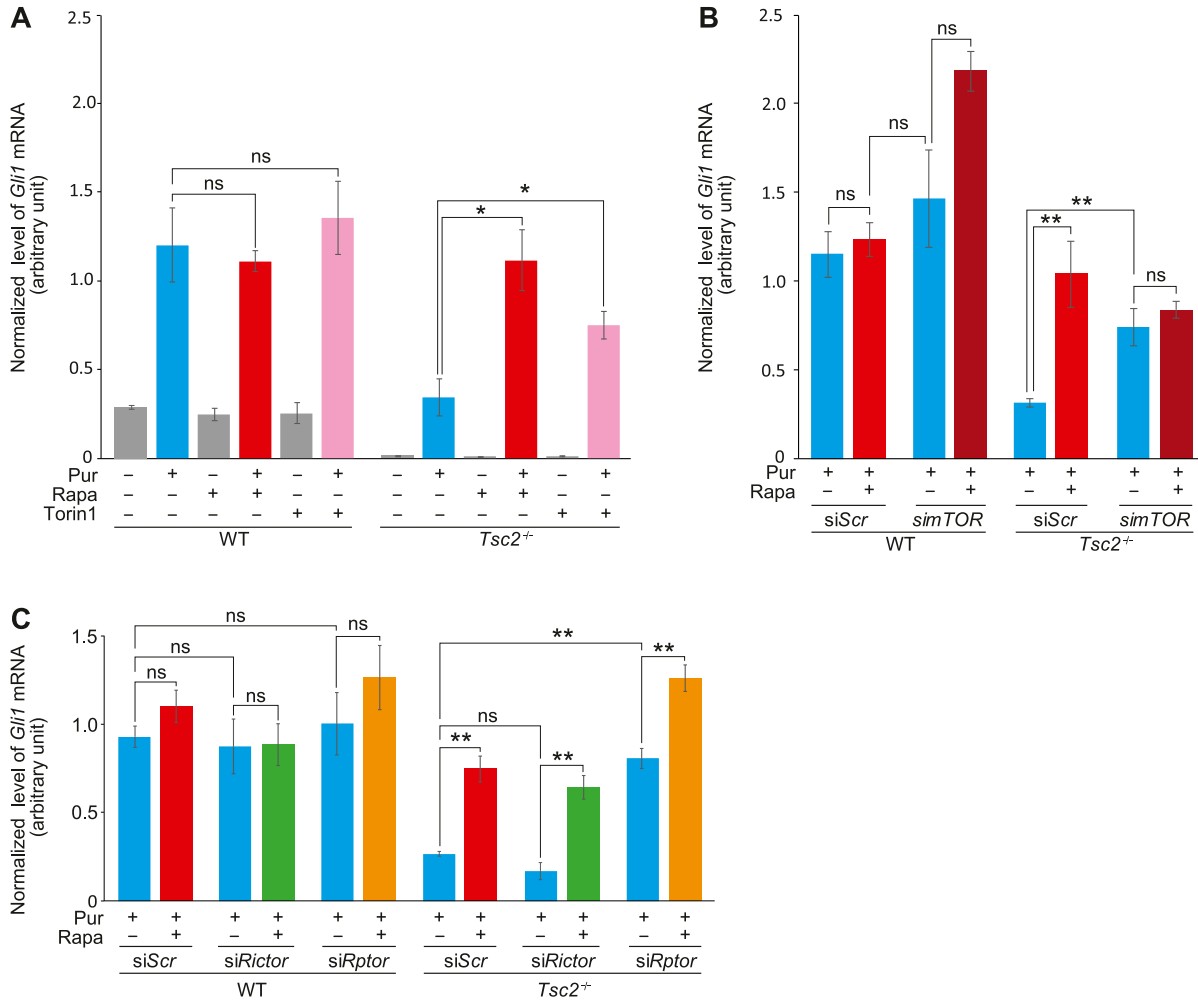

**Figure 3. mTORC1, but not mTORC2, is responsible for impaired Hh signaling in $Tsc2^{-/-}$ MEFs.**
**(A)** Inhibition of mTOR by Torin 1 rescued Hh signaling (Pur) in $Tsc2^{-/-}$ MEFs. Hh signaling was measured in the presence and absence of Torin1 and as a positive control, Rapa. WT MEFs were treated in parallel (n = 3). **(B)** mTOR regulates Hh signaling. Hh signaling (Pur) was investigated in $Tsc2^{-/-}$ MEFs after transfection with siRNA against *mTOR* or with a Scramble sequence (Scr) in the presence or absence of Rapa. WT MEFs were treated in parallel. Down-regulation of *mTOR* mRNA was verified by RT-qPCR (see Fig S4). **(C)** Down-regulation of *Rptor* but not *Rictor* rescued Hh signaling in $Tsc2^{-/-}$ MEFs. Hh signaling (Pur) was investigated after transfection with siRNA against *Rictor*, *Rptor*, or a Scramble sequence (*Scr*) in the presence or absence of Rapa (n ≥ 3). WT MEFs were treated in parallel. Down-regulation of *Rictor* and *Rptor* mRNA was verified by RT-qPCR (see Fig S5). **All**: The expression profiles of *Gli1* were normalized to *Tbp* mRNA expression (arbitrary units). Error bars represent SEM. *T* test, with significance level *$P < 0.05$, **$P < 0.01$, and ***$P < 0.001$, was used. For specific *P*-values see Table S1.

GLI2-positive cilia correlated positively with pathway activation, we would expect to see a reduced number of GLI2-positive cilia in the $Tsc2^{-/-}$ MEFs compared with WT MEFs. However, this was not the case, as more than 28% of the $Tsc2^{-/-}$ cilia were GLI2-positive, even in the absence of Pur treatment. When the cells were treated with Pur, this number increased to 71% ($P = 2.2 \times 10^{-16}$).

Treatment with Rapa, and especially in combination with Pur, reduced the level of GLI2-positive cilia in the $Tsc2^{-/-}$ MEFs, indicating that treatment with Rapa and Pur leads to the activation and nuclear translocation of GLI2 (Fig 5A). The high level of ciliary-located GLI2 combined with reduced *Gli1* expression in Pur-treated $Tsc2^{-/-}$ MEFs indicates that GLI2 is trapped inside the cilium, possibly in an inactive form.

To test whether the ciliary localization of SMO was affected in $Tsc2^{-/-}$ MEFs, the MEFs were serum-deprived for 48 h and treated

with Pur for the last 6 or 24 h to activate canonical Hh signaling. As expected, the WT MEFs displayed SMO-positive cilia in response to Pur treatment for both 6 and 24 h (Fig 5B). In contrast, in the $Tsc2^{-/-}$ MEFs, SMO was not exclusively observed in the cilium in response to Pur treatment, as it also displayed an apparent plasma membrane localization. To test whether the cellular localization was affected by Rapa treatment, we treated the cells with Pur in combination with Rapa for 24 h. Interestingly, the plasma membrane-localized SMO in the $Tsc2^{-/-}$ MEFs disappeared upon Rapa treatment, leaving a picture indistinguishable from the WT with only SMO located in the primary cilia (Fig 5B). Rapa had no effect on SMO localization in the WT MEFs (Fig 5B).

To verify the dependence of mTORC1 on SMO localization, we treated the $Tsc2^{-/-}$ MEFs with siRNA targeting the mTORC1 component RPTOR. Indeed, down-regulation of *Rptor* resulted in a

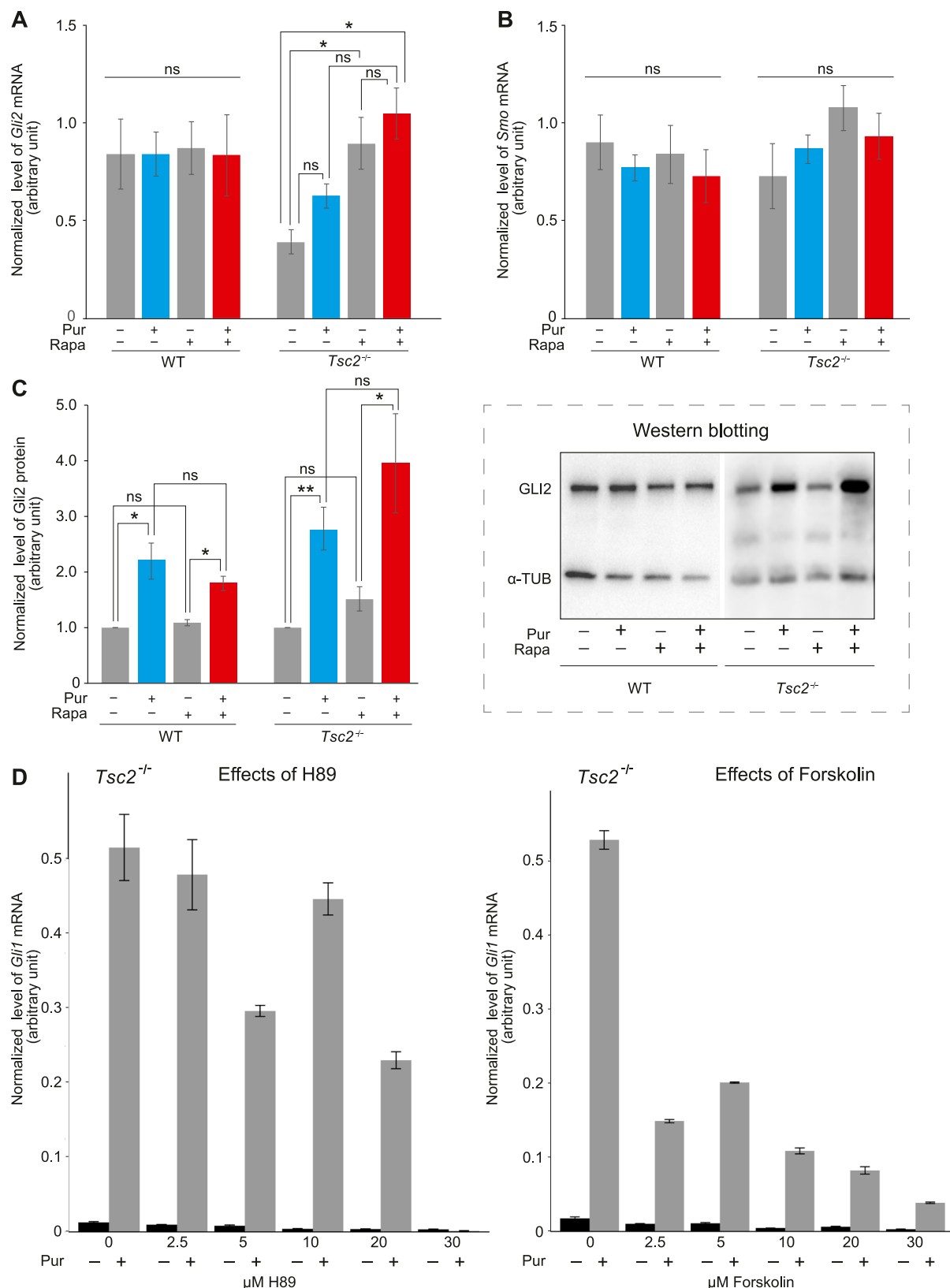

**Figure 4. Effect of Rapa on expression of Hh-actors.**
**(A)** *Gli2* mRNA level increased significantly as an effect of Rapa treatment in *Tsc2⁻/⁻* MEFs. The expression level of *Gli2* mRNA as an effect of Pur and Rapa treatment was measured in *Tsc2⁻/⁻* and WT MEFs. The expression profile of the target gene *Gli2* was normalized to *Tbp* mRNA expression. No significant differences were obtained for the

reduced level of plasma membrane-bound SMO in response to Pur treatment similar to Rapa treatment (Fig 5C). In summary, these observations indicate that mTORC1 dependent dysregulated ciliary localization of SMO and GLI2 is responsible for the deficient Hh signaling in the *Tsc2*⁻/⁻ MEFs.

### Rapa mediates activation of Hh signaling upstream of GLI2

To clarify whether Rapa affects the activation of SMO and/or the activation of GLI2, we transfected *Tsc2*⁻/⁻ cells with plasmids encoding constitutively active human GLI2 (GLI2-ΔN) or constitutively active murine SMO (SmoA1-myc) proteins. As controls, cells were transfected with plasmids encoding murine GLI2-WT or SMO-WT proteins. Transfection with GLI2-ΔN and SmoA1 both increased the level of *Gli1* mRNA, however, whereas Rapa treatment did not affect the expression of *Gli1* mRNA in cells transfected with GLI2-ΔN, Rapa treatment led to further increased expression of *Gli1* mRNA in *Tsc2*⁻/⁻ cells transfected with SmoA1-myc. These results indicate that Rapa activates Hh-signaling downstream of SMO activation but upstream of GLI2 activation. No effect of Rapa was observed in WT cells (Fig 6).

To determine whether stabilization of GLI2 takes place before SMO activation we measured the amount of GLI2 protein before and after transfection with SmoA1-myc. As seen in Fig 7, transfection with SmoA1-myc did not affect the amount of GLI2 protein, indicating that stabilization of GLI2 took place upstream of SMO activation.

## Discussion

It is well documented that defective *TSC2* leads to hyperactivation of mTORC1, but the impact of constitutive activation of mTORC1 on Hh signaling, besides our earlier brief description of this (10), has to our knowledge not been reported previously. Here we demonstrated that continuous activation of mTORC1 in mouse *Tsc2*⁻/⁻ MEFs led to reduced SMO-dependent Hh signaling, measured as a decreased expression level of *Gli1* mRNA. The negative effect of increased mTORC1 activity on Hh signaling was proven by demonstrating recovery of Hh signaling as an effect of mTORC1 inactivation.

Back in 2006, it was demonstrated that SHh signaling regulates GLI2 transcriptional activity by suppressing its processing and degradation (17). In fact, we observed an increased accumulation of GLI2 protein as a result of Pur stimulation. Despite this

accumulation of GLI2, recovery of Hh-signaling activity was only obtained in the *Tsc2*⁻/⁻ MEFs if Pur stimulation was accompanied by mTORC1 inhibition.

It has previously been shown that in the primary cilium, activated SMO transmits information to GLI2, affecting GLI2's ciliary and nuclear trafficking, leading to transcriptional activation and increased expression of Hh-target genes (12). How SMO transmits this information to GLI2 is not fully understood. In WT cells, we observed that SMO and GLI2 accumulate in the ciliary membrane upon Hh signaling; however, a different picture was observed in the *Tsc2*⁻/⁻ cells, where, even in the absence of Hh-ligand, a high fraction of the cilia contained GLI2. Upon Hh stimulation, the fraction of GLI2-containing cilia increased further, as if GLI2 were trapped in the cilium. Most notable was, however, that SMO upon Hh stimulation was partly located to the plasma membrane in the *Tsc2*⁻/⁻ cells and that inhibition of mTORC1 normalized the cellular localization of SMO and reduced the fraction of GLI2-containing cilia. We found that the cellular localization of SMO and GLI2 in *Tsc2*⁻/⁻ MEFs, as an effect of mTORC1 inhibition, occurred in parallel with rescued Hh-activity.

We demonstrated that constitutively active SmoA1 and Gli2-ΔN were able to activate the Hh-pathway, as measured by increased transcription of the Hh-target gene *Gli1*. Most importantly, the transcriptional output of *Gli1* increased after mTORC1 inactivation in cells transfected with constitutively active SMO (SmoA1), whereas mTORC1 inactivation did not affect the Hh-output in cells transfected with constitutively active GLI2 (GLI2-ΔN). These results demonstrate that Rapa acts downstream of SMO activation to promote activation of GLI2.

Taken together, these results indicate that the stabilization of GLI2 alone is not sufficient to achieve its full catalytic activity. Although Pur promoted the stability of the GLI2 protein, the increased mTORC1 activity in the *Tsc2*⁻/⁻ cells impaired the communication between SMO and GLI2. Accumulation of SMO in the cilia was inhibited, preventing activation and nuclear translocation of GLI2 (GLI2-A). This indicates that a second step besides stabilization of GLI2 protein is essential for activation and that increased mTORC1 activity inhibits this step.

Previous studies have shown that accumulation of GLI2 because of Hh stimulation is a result of decreased proteolysis and not a result of increased synthesis of GLI2 (17). Unexpectedly, we demonstrated that *Tsc2*⁻/⁻ MEFS displayed a reduced level of *Gli2* mRNA, compared to WT MEFs in a mTORC1-sensitive manner, raising the possibility that mTORC1 regulates Hh signaling through multiple mechanisms, including both protein trafficking and gene expression. However, the increase in *Gli2* mRNA level, as an effect of Rapa-

---

WT cells (n = 3). **(B)** No significant effect on the *Smo* mRNA level as an effect of Rapa or Pur treatment was observed. In the same experiment as shown in (A), the expression level of *Smo* was measured. The expression profile of the target gene *Smo* was normalized to *Tbp* mRNA expression. **(C)** The amount of accumulated GLI2 protein increased significantly as an effect of Pur treatment. Cell lysates from WT and *Tsc2*⁻/⁻ MEFs were investigated by WB, using antibodies against GLI2 full-length (FL) and α-tubulin (α-TUB), as indicated. The amount of accumulated GLI2 protein was normalized to the amount of α-TUB. The amount of accumulated GLI2 protein increased significantly in both *Tsc2*⁻/⁻ and WT MEFs as an effect of Pur treatment. No significant effect of Rapa treatment was observed. No difference in GLI2 protein accumulation in Pur-stimulated *Tsc2*⁻/⁻ MEFs treated with Rapa, compared with cells only stimulated with Pur, was observed (n = 7, *P* = 0.085). **Insert**: A representative Western blotting is shown. **(D)** Inhibition of PKA did not rescue Hh signaling (Pur) in *Tsc2*⁻/⁻ MEFs. The expression level of *Gli1* mRNA was measured in *Tsc2*⁻/⁻ MEFs treated with the PKA inhibitor H89 (left) or the PKA activator Forskolin (right) in different concentrations, as indicated, in combination with Pur (n = 1). No increase in the expression level of *Gli1* mRNA was observed as an effect of PKA inhibition (H89). As expected, a decrease in expression level of *Gli1* mRNA was observed as an effect of PKA activation (Forskolin). The expression profile of the target gene *Gli2* was normalized to *Tbp* mRNA expression. Error bars represent SEM. **(A, B, C)** *T* test, with significance level *P* < 0.05, **P* < 0.01, and ***P* < 0.001, was used. For specific *P*-values see Table S1.

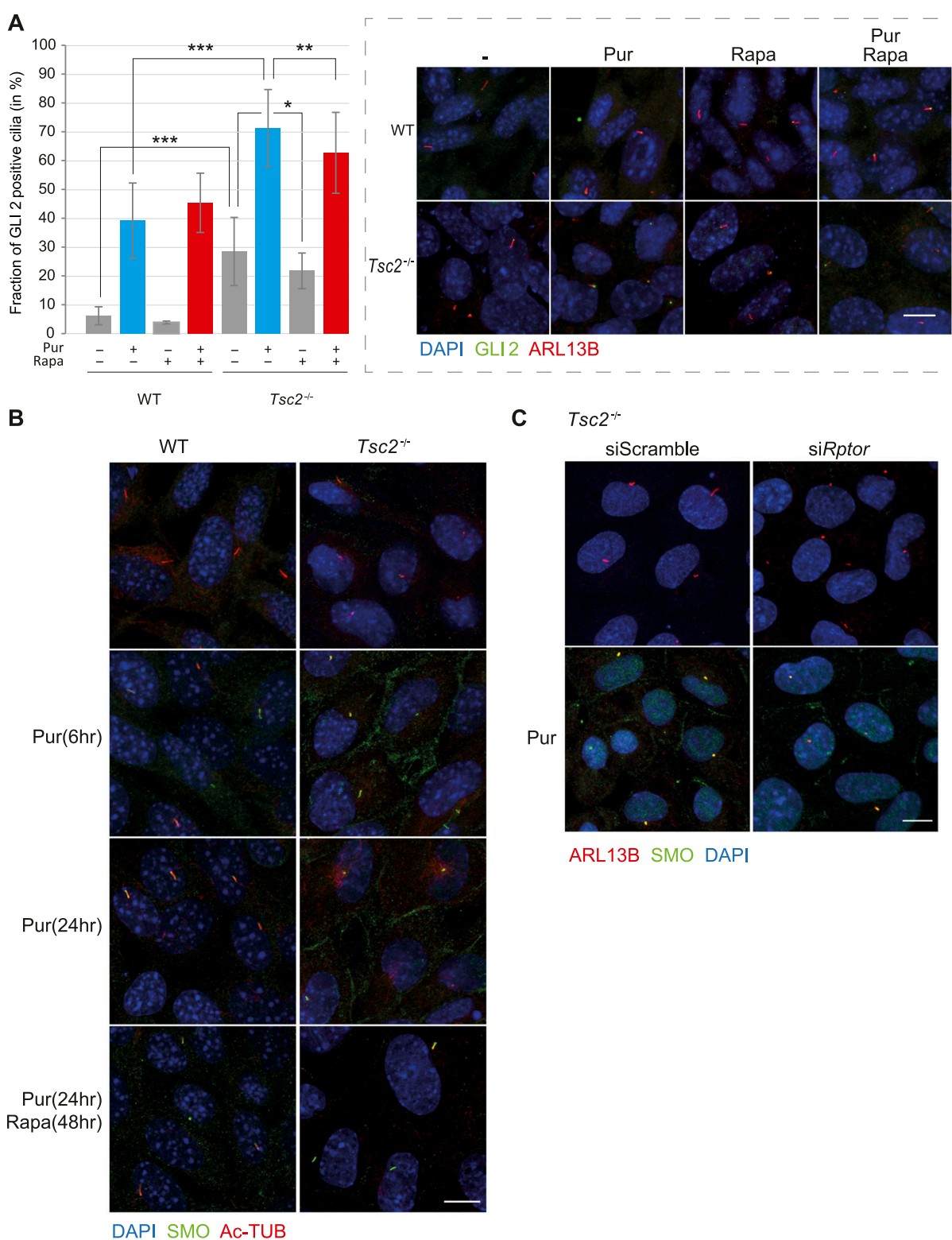

**Figure 5. GLI2 and SMO ciliary localization is affected in *Tsc2*<sup>−/−</sup> MEFs.**

**(A)** The significantly increased number of GLI2-positive cilia in *Tsc2*<sup>−/−</sup> compared with WT MEFs was normalized by Rapa treatment. Quantification of the fraction of GLI2-positive cilia of the experiment shown in Fig 4A–C, was measured as colocalization of murine GLI2 and the ciliary marker ARL13B. More than 300 cilia from three different experiments were investigated. **Insert**: A representative picture demonstrating immunofluorescence microscopy analysis of the localization of GLI2 protein, labeled with anti–GLI2 antibody (green), and primary cilia labeled with anti-ARL13B antibody (red). The nuclei were visualized by DAPI staining (blue). See also Fig S6. **(B)** Rapa rescued aberrant plasma membrane localization of SMO in *Tsc2*<sup>−/−</sup> MEFs. Cellular localization of murine SMO (green) was investigated by immunofluorescence in WT and *Tsc2*<sup>−/−</sup>

mediated inhibition of mTORC1 in $Tsc2^{-/-}$ MEFs, did not lead to an increase in accumulation of GLI2 protein. This indicates that the increased amount of $Gli2$ mRNA did not lead to increased GLI2 synthesis or that the synthesized GLI2 protein was degraded. No effect of Pur or Rapa on the expression level of $Gli2$ mRNA was observed in the WT MEFs.

In our system, we demonstrated that the effect of SMO-transmitted information to GLI2 was mTORC1 specific, as treatment with Rapa and Torin, and down-regulation of the mTORC1-specific component, RPTOR, all led to Hh-recovery in $Tsc2^{-/-}$ cells. In contrast, down-regulation of the mTORC2-specific component RICTOR had no effect on Hh-output. Whereas our results indicated an inhibitory role of mTORC1 on Hh-pathway activity, an activating role has been demonstrated for mTORC2 in a model of the malignant brain tumor glioblastoma multiforme, where it was shown that high mTORC2 activity was associated with increased expression of Hh-target genes, including $Gli1$, because of mTORC2-promoted increased stability of the GLI2 protein (33). The reason for this discrepancy is unknown but maybe because of experiments using different cell types.

Although primary cilia are essential for Hh signaling, we find it unlikely that the increased ciliation of only ~17% could explain the ~70% increased Pur-mediated $Gli1$ expression upon Rapa treatment in the $Tsc2^{-/-}$ MEFs. This is supported by our previous work (10), where we demonstrated that the frequency of ciliated cells is not significantly different in WT cells (53.6% + 10.4%) compared with $Tsc2^{-/-}$ cells (60.7% + 5.9%). Furthermore, shortened primary cilia have previously been connected with dysregulated Hh signaling (34), but although the length of the primary cilia in $Tsc2^{-/-}$ cells in fact is significantly shorter (average length of 0.74 $\mu M$), and the cilia length in $Tsc1^{-/-}$ cells is significantly longer (average length of 2.98 $\mu m$) compared with WT cells (average length 2.20 $\mu M$), we found no evidence for a direct link between cilia length and regulation of Hh signaling (10). In contrast, we paradoxically observed that Rap normalized the cilia length in $Tsc1^{-/-}$ cells but not the Hh signaling, whereas Rapa increased the Hh signaling in $Tsc2^{-/-}$ cells but not the cilia length as they were unaffected or even further reduced in length (10).

Increased mTORC1 and Hh-pathway activity have both been shown to be involved in the progression and metastasis of several cancer types (35). Analogs of Rapa such as everolimus are used as therapeutic drugs to treat patients with TSC to inhibit the growth of kidney and brain tumors, and topical mTOR inhibitors have been shown to be effective in treating skin abnormalities (36). Most of the patients (~79%) have TSC as the result of variants in $TSC2$ (37), and most studies do not discriminate between patients with $TSC1$ and $TSC2$ variants when assessing treatment efficacy. One study, which discriminated between the two genes, showed no difference between the two groups in the response to everolimus treatment of SEGAs and angiomyolipoma; however, the conclusions were based on only 13 patients with $TSC1$ variants and 84 patients with $TSC2$

variants (38). Because our study demonstrates differences in crosstalk between TSC and Hh signaling, and the effect of mTOR inhibitors, depending on the absence of either $Tsc1$ or $Tsc2$, it cannot be ruled out that the effect of treatment at least partly depends on the genetic background. We observed that Rapa treatment of $Tsc2$ null cells did lead to an increase in the Hh-signaling pathway, whereas Rapa did not lead to any increased activity of the Hh signaling in $Tsc2$ null MEFs (10). Because increased Hh-activity could stimulate the growth of tumors, analogs of Rapa might not be the optimal treatment of tumors in patients with defective $TSC2$. The effects of different drugs, including monitoring the growth of tumors, should be evaluated depending on the genetic background.

In conclusion, we have demonstrated that loss of $Tsc2$ causes impaired SMO-dependent Hh signaling through hyperactivation of mTORC1. We observed that both SMO and GLI2 had abnormal ciliary localization, which is a possible explanation for the impaired Hh signaling in $Tsc2^{-/-}$ MEFs. Furthermore, we showed that mTORC1 regulates Hh signaling at a point after SMO is activated, but before GLI2 is activated. Finally, our results indicated that the activation of GLI2 is an independent process that occurs after the stabilization of GLI2-FL, verifying a two-step activation process for GLI2.

Many questions regarding the crosstalk between Hh signaling and mTOR remain to be answered. Future studies addressing these questions will contribute to a better understanding of the molecular mechanisms involved in disease progression, revealing potential new opportunities for therapy.

# Materials and Methods

### Cell cultures and reagents

WT and $Tsc2^{-/-}$ MEF cells were obtained as a kind gift from D Kwiatkowski, Harvard University, Boston, MA, USA. Cells were cultured in DMEM supplemented with GlutaMAX, 10% FBS, and 1% penicillin–streptomycin (complete medium). The cells were grown in a 5% $CO_2$ incubator at 37°C.

For all experiments, the cells were grown under serum-deprived conditions unless otherwise specified. To induce ciliary formation, cells were serum deprived (0.5% FBS) for 48 h. Cells were treated with 25 nM Rapa (#9904; Cell Signaling Technology) or 25 nM Torin1 (475991; Sigma-Aldrich) for 48 h to inhibit mTORC1 activity. To activate Hh signaling, cells were treated with 5 $\mu$M purmorphamine (#sc-202785; Santa Cruz biotechnologies) or sonic hedgehog conditioned medium, which contains a freely diffusible form of sonic hedgehog prepared as described previously (39) for 24 h (the last 24 h of the 48 h of serum deprivation). To inhibit Hh signaling, cells were treated with 10 $\mu$M cyclopamine (15484109; Thermo Fisher Scientific) for 24 h (the last 24 h of the 48 h of serum deprivation). To

---

MEFs as an effect of Pur and Rapa treatment. Pur was administered to the medium 6 or 24 h before fixation. Primary cilia were labeled with anti-ARL13B antibody (red). The nuclei were visualized by DAPI staining (blue). The specificity of the Smo antibody was confirmed by down-regulation of $Smo$ using siRNA (Fig S3). **(C)** Down-regulation of $Rptor$ rescued aberrant plasma membrane localization of SMO similarly to Rapa. $Tsc2^{-/-}$ MEFs were transfected with siRNA against $Rptor$ or a scramble sequence before Hh stimulation for 24 h and immunofluorescence microscopy analysis of SMO (green), cilia (ARL13B, red), nuclei (blue). Down-regulation of $Rptor$ mRNA was verified by RT-qPCR (see Fig S7). Scale bar = 10 $\mu M$.

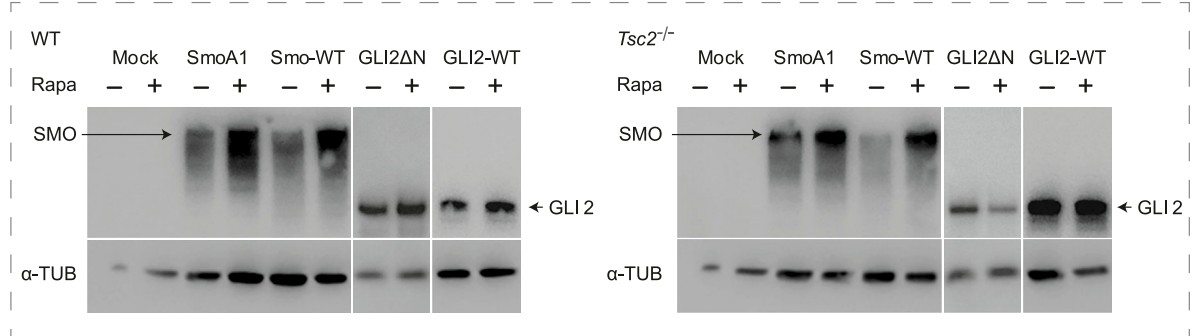

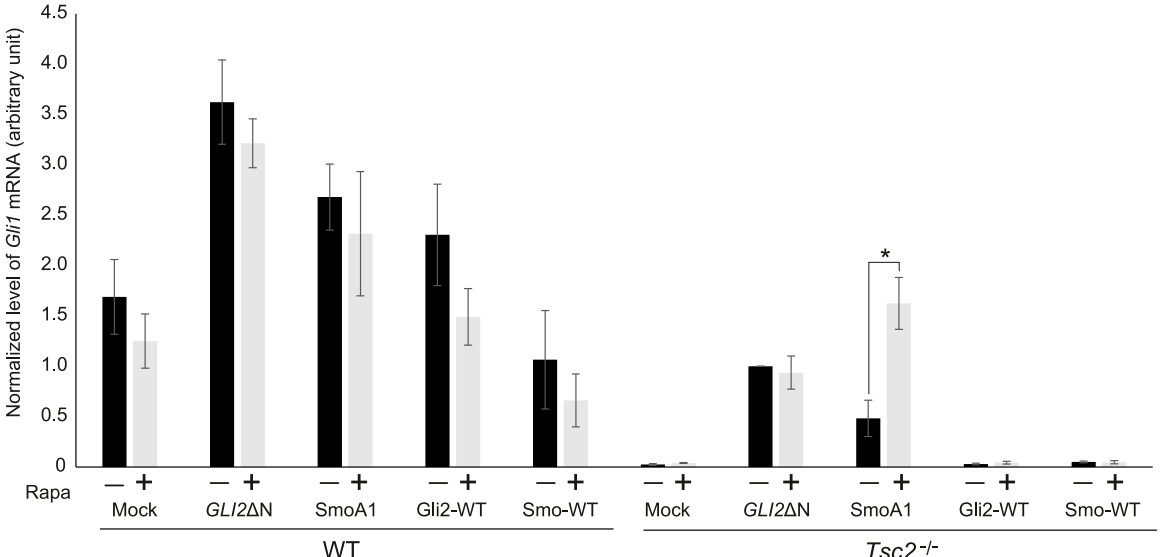

**Figure 6.  mTORC1 affects Hh-signaling downstream of SMO activation but upstream of GLI2 activation and nuclear translocation.**
WT and *Tsc2*⁻/⁻ MEFs were transfected with plasmids expressing constitutively active human GLI2 (GLI2-ΔN) or constitutively active mouse SMO (SmoA1). As controls, cells were transfected with plasmids expressing WT human GLI2 (GLI2-WT), WT mouse SMO (Smo-WT), or with an empty plasmid (Mock). After transfection, *Gli1* mRNA expression, as an indicator of Hh activation, was investigated in the presence or absence of Rapa. The expression profiles of the target gene *Gli1* were normalized to *Tbp* mRNA expression. A significant induction of *Gli1* mRNA expression was observed in *Tsc2*⁻/⁻ MEFs transfected with SmoA as an effect of Rapa treatment, whereas no effect on *Gli1* mRNA expression was observed in *Tsc2*⁻/⁻ MEFs transfected with GLI2-ΔN as an effect of Rapa treatment. No significant differences were obtained for the WT MEFs as an effect of Rapa on cells transfected with GLI2-ΔN or SmoA1 (n = 4). **Insert**: Verification of plasmid expression was performed by WB analysis of cell lysates from WT and *Tsc2*⁻/⁻ MEFs, using antibodies against human GLI2, Myc, (Myc-tag, GLI2-ΔN is myc-tagged), mouse SMO and α-tubulin as indicated.

induce autophagy, cells were treated with 10 mM Trehalose (PHR1344; Sigma-Aldrich) for 48 h.

The effect of PKA on Hh signaling was analyzed by incubation of the cells with the PKA inhibitor H89 (dihydrochloride, 371963-M; Merck) or the PKA activator Forskolin (F3917; Sigma-Aldrich) in different concentrations as indicated. The cells were cultured in serum-deprived medium (0.5% FCS) for 48 h in the presence of purmorphamine (Pur) and H89 or Forskolin for the 24 last h before mRNA harvesting.

### IFM and imaging analysis

Approximately $0.5 \times 10^6$ cells were seeded per well in a six-well plate with glass coverslips. Cells were incubated in a complete medium for 24 h before cilia induction for 48 h. Coverslips were washed three times in ice-cold PBS. They were fixed in 4% PFA for 15 min, then washed three times in PBS, permeabilized with 1% Triton X-100 in PBS for 15 min, and incubated in blocking solution (PBS containing 1% Triton X-100 and 3% BSA) for 30 min. The cells were incubated with primary antibodies overnight at 4°C and washed three times for 5 min in a blocking solution. The cells were then incubated with secondary antibodies for 45 min at RT, followed by washing for 5 min in blocking solution and then incubated with 0.5 µg/ml DAPI for 30 s to stain DNA. Unbound antibody and DAPI were removed through three additional washes in PBS, and the coverslips were then mounted with an anti-fading mounting gel containing N-propyl gallate (#P3130; Sigma-Aldrich). All procedures were performed at RT. Cells were analyzed by confocal microscopy using an Olympus Fluoview 1000 FV (Olympus). Adjustments to brightness and contrast were minimal and applied to the whole image. Z-stacked images were Z-projected and Fiji software (http://fiji.sc) was used to analyze images.

Primary antibodies used for IFM were as follows (dilutions, vendor, and catalog number in parenthesis): rabbit anti-phospho-S6, S235/236 ribosomal protein (1:1,000, # 4858; Cell Signaling), rabbit anti-S6 ribosomal protein (1:1,000, #5610; Cell Signaling),

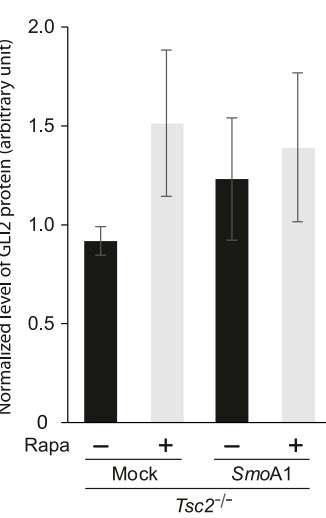
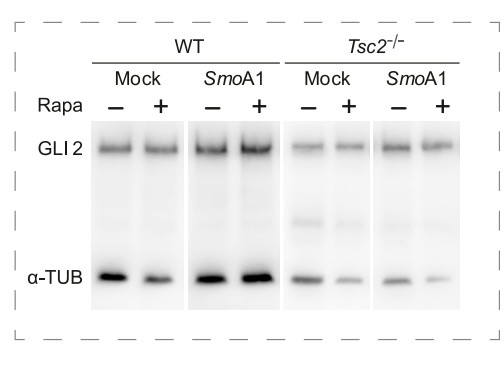

**Figure 7. Stabilization of GLI2 protein takes place upstream of SMO activation.**
WT and $Tsc2^{-/-}$ MEFs were transfected with plasmids expressing constitutively active mouse SMO (SmoA1). As controls, cells were transfected with an empty plasmid (Mock). After transfection, cells were incubated in the presence or absence of Rapa. Cell lysates from WT and $Tsc2^{-/-}$ MEFs were investigated by WB, using antibodies against GLI2 full-length (FL) and $\alpha$-tubulin, as indicated. The amount of accumulated GLI2 protein was normalized to the amount of $\alpha$-tubulin. No effect on the amount of accumulated GLI2 protein could be observed as an effect of transfection with SmoA1 or Rapa treatment (n = 4). **Insert**: A representative Western blotting is shown.

mouse anti-acetylated $\alpha$-tubulin (1:2,000, #T6793; Sigma-Aldrich), rabbit anti-Smoothened (1:1,000, ab38686; Abcam), mouse anti-Arl13b (1:1,000, #ab136648; Abcam), goat anti-Gli2 (1:1,000, AF3635; R and D Systems), rabbit anti-TSC2 (1:1,000, D93F12; Cell Signaling). Secondary antibodies used were Alexa fluor 546 donkey anti-mouse IgG (1:1,000, #A10036; Life Technologies), Alexa fluor 488 goat anti-rabbit IgG (1:1,000, #A11008; Life Technologies), and Alexa fluor 488 donkey anti-goat IgG (1:1,000, #A11055; Life Technologies).

### SDS–PAGE and Western blot analysis

The cells were lysed with ice-cold RIPA lysis buffer (150 mM sodium chloride, 1% Triton X-100, 0.5% sodium deoxycholate, 0.1% sodium dodecyl sulphate, 50 mM Tris pH 8) with the addition of 1× complete protease inhibitor cocktail tablets (#11697498001; Roche), vortexed for 10 s and left on ice for 30 min followed by centrifugation at 16,000$g$ at 4°C for 10 min. The supernatant was used for protein analysis. Total protein (20 $\mu$g) was separated by SDS–PAGE and transferred to nitrocellulose membranes. The membranes were incubated with blocking solution (100 mM Tris pH 7.6, 0.1% Tween 20, 5% dry milk) for 30 min at RT, before applying primary antibodies overnight at 4°C. Subsequently, the membranes were washed three times in TBS-T (100 mM Tris pH 7.6, 0.1% Tween 20), followed by incubation with HRP-conjugated secondary antibodies. Chemiluminescence was detected and digitally developed with G:Box Chemi XX6 (Syngene) using SuperSignal West Dura (#34076; Thermo Fisher Scientific). When needed, the membranes were stripped by two rinses in stripping solution (1% Tween 20, 0.1% SDS, 1.5% glycine, pH 2.2) followed by two rinses in PBS and TBS-T. Successful stripping of primary antibodies was confirmed by incubation with secondary antibodies followed by detection of chemiluminescence. Images were analyzed with Fiji software. The following primary antibodies were used: rabbit anti-TSC2 (1:1,000, D93F12; Cell

Signaling), murine anti-Tubulin (1:2,000, T6199; Sigma-Aldrich) goat anti-GLI2 (1:1,000, AF3635; Novus Biologicals, detect only murine GLI2), murine anti-MYC (1:1,000, 9B11; Cell Signaling), murine anti-SMO (1:1,000, E5 sc-166685; Santa Cruz), xx anti FLAG-tag. Secondary antibodies used were HRP-conjugated goat anti-mouse IgG (1:2,000, P0447; DAKO), HRP-conjugated swine anti-rabbit IgG (1:2,000, P0399; DAKO), HRP-conjugated rabbit anti-goat IgG (1:2,000, P0160; DAKO), and HRP-conjugated rabbit anti-mouse IgG (1:2,000, P0161; DAKO).

### Plasmid transfection and siRNA-mediated gene knockdown

Transfection of cells was achieved using JetPRIME (Polyplus transfection) following the protocol provided by the manufacturer. For transfection of cells plated in 12 well dishes, 1.5 $\mu$g plasmid combined with 100 $\mu$l Jetprime buffer, 3 $\mu$l Jetprime reagent, and 1 ml complete medium were added to the cells, except for pTSC2 transfection where 0.25–0.5 $\mu$g plasmid was used. The following plasmids were used: pTSC2 (a kind gift from Mark Nellist, Erasmus, MC, Rotterdam), GLI2xFlag3 (humane GLI2-WT, #84920; Addgene), GLI2deltaN (humane GLI2-ΔN, #17649; Addgene), pGenSmo (murine

**Table 1. TaqMan probes and siRNA used in the study.**

| Gene (murine) | siRNA | TaqMan probe |
|---|---|---|
| *Rictor* | SI00836640 (4) | Mm01307318_m1 |
| *Rptor* | SI00850066 (2) | Mm01242613_m1 |
| *Gli1* | | Mm00494654_m1 |
| *Smo* | SI01426838 (2) | Mm01162710_m1 |
| *Tbp* | | Mm00446971_m1 |
| *mTOR* (*FRAP1*) | SI01005690 (2) | Mm00444968_m1 |
| *Gli2* | | Mm0123111_m1 |
| *Tsc2* | SI01457106 | Mm00442004_m1 |

Smo-WT, #37673; Addgene), and pGEN-mSMOA (murine SmoA1, #37674; Addgene).

RNA silencing was achieved using DharmaFECT1 transfection reagent (Dharmacon), combined with siRNA at a final concentration of 25 nM following the protocol provided by the manufacturer. A scramble sequence was included as control. All siRNAs used were from QIAGEN (see Table 1).

### Quantitative RT–PCR analysis, expression profiles

For cDNA preparation, total RNA was extracted from the cell cultures using the GeneJet RNA purification kit (K0731; Thermo Fisher Scientific) following the instructions provided by the manufacturer. The RNA concentration was determined using an Epoch Spectrophotometer (BioTek). When needed, DNaseI (18068015; Invitrogen) was used for the degradation of plasmid or nuclear DNA present in RNA samples. Reverse transcription was carried out using the High-Capacity cDNA Reverse Transcription Kit (#4368814; Thermo Fisher Scientific). Standard qPCR was accomplished using predesigned TaqMan probes (Thermo Fisher Scientific) targeting the gene of interest combined with TaqMan Universal Master Mix (4304437; Thermo Fisher Scientific).

The measurements were performed on three independent experiments, and from each experiment, the RNA samples were analyzed in triplicate, through amplification with the TaqMan Gene Expression Master Mix (Thermo Fisher Scientific) on a 7500 Real-Time PCR system (Applied Biosystems, Thermo Fisher Scientific). The expression levels were normalized to the level of TATA-binding protein (TBP) in paralleled samples. Expression levels were evaluated through either a relative standard curve or the ΔΔCt method. Where a relative standard curve was used, the relative mean was calculated by dividing the mean expression level of the target gene with the mean expression level of the housekeeping gene (Tbp) in the sample. To calculate SD, a coefficient of variation was first calculated for both the target gene and the housekeeping gene, calculated as SD/mean (CV1 and CV2). Then the SD of the relative mean was calculated as $(CV1^2+CV2^2)^{0.5}*$relative mean.

The $C$t value is the cycle number at which the fluorescence generated within a reaction crosses the threshold line. $C$t levels are inversely proportional to the amount of target in the sample, i.e., the lower the $C$t level the greater the amount of target. $\Delta C$t is the difference in $C$t values for the target gene and the housekeeping gene ($Tbp$) for a given sample (normalized values), and $\Delta\Delta C$t is the difference in $\Delta C$t values between treated and untreated samples (40). The negative value $-\Delta\Delta C$t, is used as the exponent of 2 in the equation $2^{-\Delta\Delta Ct}$ and represents the difference in normalized number of $C$t values. The exponent 2 is used based on the assumption that each cycle doubles the amount of product.

### Statistical analysis

Quantitative results represent the mean of at least three independent experiments, if not specified otherwise. Error bars represent SEM. $P$-values were calculated using $t$ test (quantitative data) or Fisher's exact test (categorical data), if not specified otherwise. $*P < 0.05$, $**P < 0.01$, $***P < 0.001$. To correct for multiple testing Bonferroni adjustment was used.

## Data Availability

All relevant data of this study are available within the article and its Supplementary Information files or from corresponding author on request.

## Supplementary Information

## Acknowledgements

We thank D Kwiatkowski, Harvard University, Boston, for the MEF cells. We also thank Katia Stæhr Vinding for excellent technical support and Jette Bune Rasmussen for assistance with generating the figures. The study was supported by grants from the Independent Research Fund Denmark (12-127196 #0602-02725B), Familien Hede Nielsen's Foundation, Einar Willumsen's Mindelegat, and Dagmar Marshall's foundation.

### Author Contributions

LJ Larsen: conceptualization, data curation, validation, investigation, methodology, and writing—original draft.
E Østergaard: conceptualization, funding acquisition and writing—review and editing.
LB Møller: conceptualization, data curation, supervision, funding acquisition, investigation, methodology, project administration, and writing—original draft, review, and editing.

### Conflict of Interest Statement

The authors declare that they have no conflict of interest.

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
