## [Reviewer comments · Life Science Alliance]

mTORC1 hampers Hedgehog signaling in Tsc2 deficient cells

Lasse J Larsen, Elsebet Østergaard and Lisbeth B Møller

DOI: 10.26508/lsa.202302419

Corresponding author(s): Dr. Lisbeth Birk Møller (Rigshospitalet)

Review timeline:

Submission Date:	2023-10-05
Editorial Decision:	2023-11-13
Appeal Requested:	2024-07-12
Editorial Decision:	2024-07-15
Revision Received:	2024-07-16
Editorial Decision:	2024-08-08
Revision Received:	2024-08-09
Accepted:	2024-08-16

Scientific Editor: Eric Sawey

Transaction Report:

Re: Life Science Alliance manuscript #LSA-2023-02419-T

Dr. Lisbeth Birk Møller
Rigshospitalet
Genetic department
Gl landevej 7
Glostrup, Denmark 2600
Denmark

Dear Dr. Møller,

Thank you for submitting your manuscript entitled "Hyperactive mTORC1 affects cellular localization of Smoothed and inhibits GLI2 activation". The manuscript has been evaluated by expert reviewers, whose reports are appended below. Unfortunately, after an assessment of the reviewer feedback, our editorial decision is against publication in Life Science Alliance.

Although your manuscript is intriguing, I feel that the points raised by the reviewers are more substantial than can be addressed in a typical revision period. If you wish to expedite publication of the current data, it may be best to pursue publication at another journal.

Given the interest in the topic, I would be open to re-submission to Life Science Alliance of a significantly revised and extended manuscript that fully addresses the reviewers' concerns and is subject to further peer review. If you would like to resubmit this work to Life Science Alliance, you may submit an appeal directly through our manuscript submission system. Please note that priority and novelty would be reassessed at re-submission.

Regardless of how you choose to proceed, we hope that the comments below will prove constructive as your work progresses.

Thank you for thinking of Life Science Alliance as an appropriate place to publish your work.

Sincerely,

Reviewer #1 (Comments to the Authors (Required)):

The paper by Larsen et al "Hyperactive mTORC1 affects cellular localization of Smoothed and inhibits GLI2 activation" describes the dissection of the pathway that leads to impaired Hh- signaling in TSC2 deficient mouse cells (MEFs from knock out mice).

The authors use many approaches and some of them are elegant. However, some of the experiments seem to be replica of previous published experiments by the same authors: E.g. reference 8: figure 2 in this paper seems very much alike figure 4A in reference 8. Now, this may be absolutely OK and give the reader a context, but it also stands as new knowledge which it is not. The authors could refer to the previous experiments

Overall, the paper is challenging to read, and in its present form, reserved a limited group of readers. I believe that figure 8 could be expanded so that it would lead back to many of the experiments done. A more graphical view of the different attack points between Tsc2 and Hh-signaling would be helpful. A view that also includes siRNA experiments.

The authors have a number of statements regarding the implications of their work. E.g. that genetic status should be used when evaluating TSC treatment using mTor inhibitors. I may be mistaken, but, I think that a failure or difference in response between TSC1 or TSC2 patients would have been picked up by now. This might point to a weakness in using mouse cells in this study instead of human cells. Human TSC2-cells could be made quite easily using CRISPR technology.

The paper is not always clear regarding reference to previous or present experiments. This is particularly problematic in the introduction. For example page 4, section 5 of the introduction starting with "Canonical Hh-signaling....." : the sentence starting with: "In the presence of Hh-ligand, PTCH1 is displaced away from the primary cilium....." Please give reference to this.

There are a few grammatical glitches. Eg: last section of the introduction. " whether either Tsc1 or Tsc2 is absent". Either may not be correct and unnecessary.

Nomenclature needs some attention. Gli1 is sometimes GLI1 in the figures. Please adhere to convention regarding mouse genes and proteins.

Reviewer #2 (Comments to the Authors (Required)):

Manuscript LSA-2023-02419-T, 'Hyperactive mTORC1 affects cellular localization of Smoothened and inhibits GLI2 activation' by Larsen, Ostergaard and Moller describes a series of experiments investigating Hedgehog (Hh) signaling and primary cilium biogenesis in Tsc2 KO mouse embryonal fibroblasts (MEFs). The authors show that Tsc2 KO affects Hh signaling, and that this is likely due to defective mTOR signaling. Specifically, Tsc2 KO MEFs show reduced Gli1 mRNA expression that can be rescued by re-expression of Tsc2 or by down-regulation of mTORC1. The manuscript advances the field by showing that Tsc2 KO, like Tsc1 KO, affects Hh signaling in MEFs. The authors conclude that Tsc2 and Tsc1 KO have distinct effects on GLI2 expression and localization and speculate that this might affect the response of individuals with TSC to therapy.

A considerable number of experiments have been performed, and some convincing differences between wild-type (WT) and Tsc2 KO MEFs have been identified. I am sorry, but it was difficult for me to follow at times (my lack of knowledge). However, my main concern is that the observed differences might simply be a secondary effect of Tsc2-dependent (but maybe also Tsc2 independent?) differences in the growth, division and cell cycle of the different cell-lines used. My simple (naive) hypothesis would be that if the cells are growing more quickly, there is less time to re-construct the primary cilium, possibly explaining the observed differences in Gli1 expression. Are there differences between WT and Tsc2 KO MEFs in the number and structure of the primary cilia? This information would have been useful (or maybe I missed it?).

I realize it is outside the scope of the present manuscript, but confirmation of the described effects on Hh signaling in independently-derived Tsc2 (and Tsc1!) KO cell-lines would be reassuring.

I had some specific comments on the manuscript:

1. The authors use RT-PCR to show that the expression of Gli1 is reduced in Tsc2 KO MEFs (Figure 1). It was not clear to me if the response to Shh and/or purmorphamine was attenuated. What was the fold induction? Possibly this is not clear because I am not sure what the y-axis Gli1 mRNA/Tbp mRNA represents in Figure 1. Does this indicate the difference in signals from an end-point PCR, or the deltaCt value from a Q-RT-PCR? It would also have been helpful to include Tsc1 KO MEFs, and ideally another siRNA to knock-down another gene known to be essential for primary cilium biogenesis and Hh signaling. Such experiments would help show how large the effects of loss of Tsc2 are on Gli1 expression. I found this difficult to understand from the data, as presented.

2. The authors compare the number of cilia with induction of Gli1 expression. Why can't the increased ciliation explain the increased Gli1 expression? This was not clear to me.
3. To investigate whether mTORC1 and/or mTORC2 play a role in the observed increase in Hh signaling, the authors compare the effects of mTOR inhibition with rapamycin and torin 1, and siRNA-mediated knock-down of mTor, Rictor, Raptor, S6k1 and 4e-bp1. I think it might be helpful to combine Figure 3B and 3C with Figure 4A and 4B. These graphs all show the effect of siRNA-mediated knock-down in WT and Tsc2 KO MEFs. Being able to directly compare the effects of the different treatments would help show which genes are important for the induction of Gli1 expression.
4. Is the observed additive effect of rapamycin treatment with Raptor knock-down not just simply due to a more complete inhibition of mTOR activity?
5. 'No effect of Gli2 mRNA expression of neither Pur nor rapamycin was observed in the WT MEFs'. I am not sure what the authors mean with this statement, but I think that they should avoid double negative statements: no effect of neither/nor, implies an effect of either or both, right?
6. In Figure 5C the authors show the effect of purmorphamine and rapamycin on Gli2 protein expression. It would be helpful to indicate the comparison mentioned in the text in the figure. Did the authors correct for multiple testing?
7. What are the units of Gli1 mRNA expression in Figure 5D?
8. For comparison and for completeness, please include the blots for Gli2-WT and Smo-WT, for both the WT and Tsc2 KO cells in Figure 7. To convincingly demonstrate differences between Tsc2 and Tsc1 KO with respect to GLI2, I think that it would be necessary to compare both cell-lines in the experiments shown in Figures 5 and 6.
9. Please confirm that for the quantification of the RT-PCR data, 3 independent experiments were performed, and from each experiment the RNA samples were analysed in triplicate. Is that correct?

Reviewer #3 (Comments to the Authors (Required)):

Hedgehog signaling and mTORC pathways are known to interact with each other in cancer, but the molecular mechanism remains unknown. Here in this study, the authors attempted to map the interaction between Hedgehog signaling and mTORC1 signaling in Tsc knockout cells and found that mTORC1 hyperactivation inhibits Hedgehog signaling. mTORC1 signaling seems to affect both ciliation and activation of GLI transcription factors. These findings provide key insights into interactions between the two pathways and how they contribute to the phenotype seen in TSC patients.

The findings from this study are significant, but can benefit from some more detailed characterization. Ciliation has a major impact on Hedgehog pathway activation and the changes in Hedgehog signaling activity in this study are thus always complicated by different ciliation efficiency in knockout vs WT cells. Can the authors repeat the experiments measuring ciliation and Hedgehog pathway activity in wildtype cells after acute activation of mTORC1 signaling? If mTORC1 is acutely activated after ciliation, then levels of ciliation would no longer be a compounding factor for Hedgehog pathway activity, yielding more convincing results.

In addition, Smo conformation affects its own ciliary accumulation. Can Smo accumulation in the cilia be quantified? Does Tsc knockout affect ciliary accumulation of SmoA1 variant? These experiments would be important controls to map the step affected by mTORC1.

Appeal Request

12 July 2024

Dear Dr. Sawey, PhD,

The authors of manuscript #LSA-2023-02419-T have requested an appeal. Their comments are below.

Copenhagen, 11 July 2024

Dear editor,

Thank you for all the very useful comments to the previous version. We have now performed some additional experiments and have in the following responded to all of them one by one.

We hereby submit the revised manuscript entitled "Hyperactive mTORC1 affects cellular localization of Smoothed and inhibits GLI2 activation" by Lasse Jonsgaard Larsen, Elsebet Østergaard, Lisbeth Birk Møller, to be considered for publication as a research paper in the international journal "Life Science Alliance"

In the present paper we have investigated the effect of Tsc2 on Hh-signaling in detail, and revealed a new, previously unknown interaction between mTORC1 and Hh-signaling. Our data indicated that reduced Hh-activation in Tsc2-null cells were due to hampered GLI2 activation and nuclear translocation, and that this might be a result of impaired ciliary location of SMO. Our results supported previous studies demonstrating that the activation and nuclear translocation of GLI2 is dependent on cilia-located activated SMO, but furthermore demonstrate a hitherto unknown impact of mTORC1 on SMO trafficking and GLI2 activation and revealed evidence for a 2-step activation process of GLI2.

Sincerely yours,

Lisbeth Birk Møller, Department of Genetics, Kennedy Center, Copenhagen University Hospital, Rigshospitalet, Glostrup, Denmark

Sincerely,

Editorial Staff

Editorial Decision on Appeal Request

15 July 2024

MS: LSA-2023-02419-T

Dr. Lisbeth Birk Møller
Rigshospitalet
Genetic department
GI landevej 7
Glostrup, Denmark 2600
Denmark

Dear Dr. Møller,

Your manuscript entitled "Hyperactive mTORC1 affects cellular localization of Smoothed and inhibits GLI2 activation" has now been reconsidered, and I am pleased to let you know that we have decided to send your revised manuscript back to the original Reviewers.

Please use the following link to submit your revised manuscript:

[link removed by Editorial Staff]

Yours sincerely,

Life Science Alliance Resubmission of LSA-2023-02419-T Dear Editor, Eric Sawey Thank you for all the very useful comments. We have performed some new experiments and have in the following responded to all of them one by one. Regards Lisbeth

----- Reviewer #1 (Comments to the Authors (Required)): The paper by Larsen et al "Hyperactive mTORC1 affects cellular localization of Smoothed and inhibits GLI2 activation" describes the dissection of the pathway that leads to impaired Hh- signaling in TSC2 deficient mouse cells (MEFs from knock out mice). The authors use many approaches and some of them are elegant. However, some of the experiments seem to be replica of previous published experiments by the same authors: E.g., reference 8: figure 2 in this paper seems very much alike figure 4A in reference 8. Now, this may be absolutely OK and give the reader a context, but it also stands as new knowledge which it is not. The authors could refer to the previous experiments.

Overall, the paper is challenging to read, and in its present for, reserved a limited group of readers. I believe that figure 8 could be expanded so that it would lead back to many of the experiments done. A more graphical view of the different attack points between Tsc2 and Hh-signaling would be helpful. A view that also includes siRNA experiments.

Respond: We have now expanded figure 8 (this figure is figure 8 in the revised version). We believe that this will help the reader.

We have also now at several points in the revised version referred more clearly to our previously work if appropriate. Furthermore, we have in general tried to improve the text to make it easier to read. In addition, we have deleted the siRNA experiments regarding siSp6 and si4EBP as they are not important for this paper.

We hope this will all together make it easier to read the paper.

The authors have a number of statements regarding the implications of their work. E.g., that genetic status should be used when evaluating TCS treatment using mTor inhibitors. I may be mistaken, but I think that a failure or difference in response between TSC1 or TSC2 patients would have been picked up by now. This might point to a weakness I using mouse cells I this study in stead of human cells. Human TSC2- cells could be made quite easily using CRISPR technology.

Respond: We believe that the difference in response between TSC1 and TSC2 patients are not clarified. We have, to make this point clearer, included this revised chapter in the new version

“Increased mTORC1 and Hh-pathway activity have both been shown to be involved in the progression and metastasis of several cancer types [35]. Analogs of Rapa such as everolimus are used as therapeutic drugs to treat patients with TSC to inhibit the growth of kidney and brain tumors, and topical mTOR inhibitors have been shown to be effective in treating skin abnormalities [36]. The majority of patients (~ 79%) have TSC as the result of variants in TSC2 [37], and most studies do not discriminate between patients with TSC1 and TSC2 variants when assessing treatment efficacy. One study which discriminate between the genotypes showed no difference between the two groups in the response to everolimus treatment of SEGAs and angiomyolipoma; however, the conclusions were based on only 13 patients with TSC1 variants and 84 patients with TSC2 variants [38]. Since our study demonstrates differences in crosstalk between TSC and Hh signaling, and the effect of mTOR inhibitors, depending on the absence of either Tsc1 or Tsc2, it cannot be ruled out that the effect of treatment at least partly depends on the genetic background. We observed that Rapa treatment of Tsc2 null cells did lead to an increase of the Hh-signaling pathway, whereas Rapa did not lead to any increased activity of the Hh-signaling in Tsc2 null MEFs [10]. Since increased Hh-activity could stimulate the growth of tumors, analogs of Rapa might not be the optimal treatment of tumors in patients with defective TSC2. The effects of different drugs, including monitoring the growth of tumors, should be evaluated depending on the genetic background.”

The paper is not always clear regarding reference to previous or present experiments. This is particularly problematic in the introduction. For example, page 4, section 5 of the introduction starting with "Canonical Hh-signaling....." : the sentence starting with: "In the presence of Hh ligand, PTCH1 is displaced away from the primary cilium....." Please give reference to this.

Respond: We have now in the revised version added several additional references.

There are a few grammatical glitches. Eg: last section of the introduction. " whether either Tsc1 or Tsc2 is absent". Either may not be correct and unnecessary.

Respond. We have now read the paper carefully, hopefully the number of mistakes is reduced.

Nomenclature needs some attention. Gli1 is sometimes GLI1 in the figures. Please adhere to convention regarding mouse genes and proteins.

Respond: We have now corrected the nomenclature.

Reviewer #2 (Comments to the Authors (Required)): Manuscript LSA-2023-02419-T, 'Hyperactive mTORC1 affects cellular localization of Smoothed and inhibits GLI2 activation' by Larsen, Ostergaard and Moller describes a series of experiments investigating Hedgehog (Hh) signaling and primary cilium biogenesis in Tsc2 KO mouse embryonal fibroblasts (MEFs). The authors show that Tsc2 KO affects Hh signaling, and that this is likely due to defective mTOR signaling. Specifically, Tsc2 KO MEFs show reduced Gli1 mRNA expression that can be rescued by re-expression of Tsc2 or by down-regulation of mTORC1. The manuscript advances the field by showing that Tsc2 KO, like Tsc1 KO, affects Hh signaling in MEFs. The authors conclude that Tsc2 and Tsc1 KO have distinct effects on GLI2 expression and localization and speculate that this might affect the response of individuals with TSC to therapy.

A considerable number of experiments have been performed, and some convincing differences between wild-type (WT) and Tsc2 KO MEFs have been identified. I am sorry, but it was difficult for me to follow at times (my lack of knowledge). However, my main concern is that the observed differences might simply be a secondary effects of Tsc2-dependent (but maybe also Tsc2 independent?) differences in the growth, division and cell cycle of the different cell-lines used. My simple (naive) hypothesis would be that if the cells are growing more quickly, there is less time to re-construct the primary cilium, possibly explaining the observed differences in Gli1 expression. Are there differences between WT and Tsc2 KO MEFs in the number and structure of the primary cilia? This information would have been useful (or maybe I missed it?).

Respond: That is an important point. We have investigated this point in our previous work (Rosengren T, Larsen LJ, Pedersen LB, et al. TSC1 and TSC2 regulate cilia length and canonical Hedgehog signaling via different mechanisms. *Cell Mol Life Sci.* 2018;75:2663–2680. PMID:29396625) but we agree that this information is usefull to have and have now included in the discussion:

“Although primary cilia are essential for Hh-signaling we find it unlikely that the increased ciliation of only ~17% could explain the ~70% increased Pur-mediated Gli1 expression upon Rapa treatment in the Tsc2^{-/-} MEFs. This is supported by our previously work [10] where we demonstrated that the frequency of ciliated cells is not significantly different in WT cells (53.6 +10.4 %) compared to Tsc2^{-/-} (60.7+5.9 %). Furthermore, shortened primary cilia have previously been connected with dysregulated Hh-signaling [34], but although, the length of the primary cilia in Tsc2^{-/-} cells, in fact is significantly shorter (average length of 0.74μM), and the cilia length in Tsc1^{-/-} cells significantly longer (average length of 2.98μm) compared to WT cells (average length 2.20μM) we found no evidence for a direct link between cilia length and regulation of Hh-signaling [10]. In contrast, we paradoxically observed that Rap normalized the cilia length in Tsc1^{-/-} cells but not the Hh-signaling whereas Rap increased the Hh-signaling in Tsc2^{-/-} cells but not the cilia length as they were unaffected or even further reduced in length [10].”

I realize it is outside the scope of the present manuscript, but confirmation of the described effects on Hh signaling in independently-derived Tsc2 (and Tsc1!) KO cell-lines would be reassuring.

Respond: We agree this is an important experiment, but this needs to be published in a new paper. We agree that this is behind the scope of this paper

I had some specific comments on the manuscript:

1. The authors use RT-PCR to show that the expression of Gli1 is reduced in Tsc2 KO MEFs (Figure 1). It was not clear to me if the response to Shh and/or purmorphamine was attenuated. What was the fold induction? Possibly this is not clear because I am not sure what the y-axis Gli1 mRNA/Tbp mRNA represents in Figure 1. Does this indicate the difference in signals from an end-point PCR, or the deltaCt value from a Q-RT-PCR? It would also have been helpful to include Tsc1 KO MEFs, and ideally another siRNA to knock-down another gene known to be essential for primary cilium biogenesis and Hh signaling. Such experiments would help show how large the effects of loss of Tsc2 are on Gli1 expression. I found this difficult to understand from the data, as presented.

Respond: That is a good point. Actually, the total amount of Gli1 mRNA was reduced in Tsc2KO cells compared to WT after Hh stimulation. But if we calculate the fold induction the fold induction was higher in the Tsc2KO cells compared to WT. We have now added this figure to supplementary and made a comment on this in the text. "Although the expression level of Gli1 was lower in the Tsc2^{-/-} cells compared with the WT cells, the SHh-induced increased expression of Gli1 mRNA in the Tsc2^{-/-} cells, verifying that the cells were Hh signal responsive. Indeed, as a result of significantly lower basal level of Gli1 mRNA in the Tsc2^{-/-} cells, compared to the WT cells (23-34-fold), the fold increase in expression of Gli1 mRNA was significantly higher in the Tsc2^{-/-} cells compared to the WT cells (7-12-fold) (Figure S1)."

However, since the interesting part of our study is that Rapa leads to increased expression of Gli1 mRNA in Tsc2 KO MEFs and not in WT, we have focused on the amount of Gli mRNA (and not the fold induction). The y-axis Gli1 mRNA/Tbp mRNA represents the deltaCt value from a Q-RT-PCR. We have now made this clearer in the figure legend and have in addition changed the legend on the Y-axis to: Normalized level of Gli1 mRNA (arbitrary unit). We have not included TSC1 KO because we have investigated this cell-line previously (ref: Rosengren T, Larsen LJ, Pedersen LB, et al. TSC1 and TSC2 regulate cilia length and canonical Hedgehog signaling via different mechanisms. Cell Mol Life Sci. 2018;75:2663–2680. PMID:29396625).

2. The authors compare the number of cilia with induction of Gli1 expression. Why can't the increased ciliation explain the increased Gli1 expression? This was not clear to me.

Respond: That is an important point. We have investigated this point in our previous work (Rosengren T, Larsen LJ, Pedersen LB, et al. TSC1 and TSC2 regulate cilia length and canonical Hedgehog signaling via different mechanisms. *Cell Mol Life Sci.* 2018;75:2663–2680. PMID:29396625) but we agree that this information is useful to have and have now included in the discussion:

“Although primary cilia are essential for Hh-signaling we find it unlikely that the increased ciliation of only ~17% could explain the ~70% increased Pur-mediated Gli1 expression upon Rapa treatment in the *Tsc2*^{-/-} MEFs. This is supported by our previously work [10] where we demonstrated that the frequency of ciliated cells is not significantly different in WT cells (53.6 +10.4 %) compared to *Tsc2*^{-/-} (60.7+5.9 %). Furthermore, shortened primary cilia have previously been connected with dysregulated Hh-signaling [34], but although, the length of the primary cilia in *Tsc2*^{-/-} cells, in fact is significantly shorter (average length of 0.74μM), and the cilia length in *Tsc1*^{-/-} cells significantly longer (average length of 2.98μm) compared to WT cells (average length 2.20μM) we found no evidence for a direct link between cilia length and regulation of Hh-signaling [10]. In contrast, we paradoxically observed that Rap normalized the cilia length in *Tsc1*^{-/-} cells but not the Hh-signaling whereas Rap increased the Hh-signaling in *Tsc2*^{-/-} cells but not the cilia length as they were unaffected or even further reduced in length [10].”

3. To investigate whether mTORC1 and/or mTORC2 play a role in the observed increase in Hh signaling, the authors compare the effects of mTOR inhibition with rapamycin and torin 1, and siRNA-mediated knock-down of mTor, Rictor, Raptor, S6k1 and 4e-bp1. I think it might be helpful to combine Figure 3B and 3C with Figure 4A and 4B. These graphs all show the effect of siRNA-mediated knock-down in WT and *Tsc2* KO MEFs. Being able to directly compare the effects of the different treatments would help show which genes are important for the induction of Gli1 expression.

Respond We have now combined figure 3 and 4 into a new figure 3. In addition, we have deleted the siRNA experiments regarding siSp6 and si4EBP as they are not important for this paper. We hope this will all together make it easier to read the paper.

4. Is the observed additive effect of rapamycin treatment with Raptor knock-down not just simply due to a more complete inhibition of mTOR activity?

Respond. Yes, we agree. We have now deleted this sentence.

5. 'No effect of Gli2 mRNA expression of neither Pur nor rapamycin was observed in the WT MEFs'. I am not sure what the authors mean with this statement, but I think that they should avoid double negative statements: no effect of neither/nor, implies an effect of either or both, right?

Respond. Thank you for this important point. We have now changed it to: No effect on Gli2 mRNA expression of Pur or Rapa was observed in the WT MEFs 6.

In Figure 5C the authors show the effect of purmorphamine and rapamycin on Gli2 protein expression. It would be helpful to indicate the comparison mentioned in the text in the figure. Did the authors correct for multiple testing?

Respond: Yes, we correct for multiple testing. In material and methods, it is now added:

“Quantitative results represent the mean of at least three independent experiments, if not specified otherwise. Error bars represent standard error of the mean (SEM). p values were calculated using Student’s t test (quantitative data) or Fisher’s exact test (categorical data), if not specified otherwise. *p < 0.05, **p < 0.01, ***p < 0.001. To correct for multiple testing Bonferroni adjustment was used.”

7. What are the units of Gli1 mRNA expression in Figure 5D?

Respond: Normalized level of Gli1 mRNA (arbitrary unit). It has now been added to the figures.

8. For comparison and for completeness, please include the blots for Gli2-WT and Smo-WT, for both the WT and Tsc2 KO cells in Figure 7.

Respond: This is now performed in the revised figure 6

To convincingly demonstrate differences between Tsc2 and Tsc1 KO with respect to GLI2, I think that it would be necessary to compare both cell-lines in the experiments shown in Figures 5 and 6.

Respond: We have not included TSC1 KO because we have previously shown GLI2 in Tsc1 and Tsc2 KO (ref: Rosengren T, Larsen LJ, Pedersen LB, et al. TSC1 and TSC2 regulate cilia length and canonical Hedgehog signaling via different mechanisms. Cell Mol Life Sci. 2018;75:2663–2680. PMID:29396625).

9. Please confirm that for the quantification of the RT-PCR data, 3 independent experiments were performed, and from each experiment the RNA samples were analysed in triplicate. Is that correct?

Respond Yes this is correct

Reviewer #3 (Comments to the Authors (Required)): Hedgehog signaling and mTORC pathways are known to interact with each other in cancer, but the molecular mechanism remains unknown. Here in this study, the authors attempted to map the interaction between Hedgehog signaling and mTORC1 signaling in Tsc knockout cells and found that mTORC1 hyperactivation inhibits Hedgehog signaling. mTORC1 signaling seems to affect both ciliation and activation of GLI transcription factors. These findings provide key insights into interactions between the two pathways and how they contribute to the phenotype seen in TSC patients.

The findings from this study are significant, but can benefit from some more detailed characterization. Ciliation has a major impact on Hedgehog pathway activation and the changes in Hedgehog signaling activity in this study are thus always complicated by different ciliation efficiency in knockout vs WT cells. Can the authors repeat the experiments measuring ciliation and Hedgehog pathway activity in wildtype cells after acute activation of mTORC1 signaling? If mTORC1 is acutely activated after ciliation, then levels of ciliation would no longer be a compounding factor for Hedgehog pathway activity, yielding more convincing results.

Respond: That is an important point. We have investigated this point in our previous work (Rosengren T, Larsen LJ, Pedersen LB, et al. TSC1 and TSC2 regulate cilia length and canonical Hedgehog signaling via different mechanisms. *Cell Mol Life Sci.* 2018;75:2663–2680. PMID:29396625) but we agree that this information is useful to have and have now included in the discussion:

“Although primary cilia are essential for Hh-signaling we find it unlikely that the increased ciliation of only ~17% could explain the ~70% increased Pur-mediated Gli1 expression upon Rapa treatment in the Tsc2^{-/-} MEFs. This is supported by our previously work [10] where we demonstrated that the frequency of ciliated cells is not significantly different in WT cells (53.6 +10.4 %) compared to Tsc2^{-/-} (60.7+5.9 %). Furthermore, shortened primary cilia have previously been connected with dysregulated Hh-signaling [34], but although, the length of the primary cilia in Tsc2^{-/-} cells, in fact is significantly shorter (average length of 0.74μM), and the cilia length in Tsc1^{-/-} cells significantly longer (average length of 2.98μm) compared to WT cells (average length 2.20μM) we found no evidence for a direct link between cilia length and regulation of Hh-signaling [10]. In contrast, we paradoxically observed that Rap normalized the cilia length in Tsc1^{-/-} cells but not the Hh-signaling whereas Rap increased the Hh-signaling in Tsc2^{-/-} cells but not the cilia length as they were unaffected or even further reduced in length [10].”

In addition, Smo conformation affects its own ciliary accumulation. Can Smo accumulation in the cilia be quantified? Does Tsc knockout affect ciliary accumulation of SmoA1 variant? These experiments would be important controls to map the step affected by mTORC1.

Respond. We agree it is an important point. However even though we have tried several times to perform this experiment we are not able to see the specific localization of the SmoA1 variant in our cells. However, to obtain more information about what is going on according to the timeline for GLI2 stabilization and SmoA1 activation we have added a new figure (Figure 7): As seen in Figure 7 transfection with SmoA1-myc did not affect the amount of GLI2 protein, verifying that stabilization of GLI2 took place upstream SMO activation.

RE: Life Science Alliance Manuscript #LSA-2023-02419-TR-A

Dr. Lisbeth Birk Møller
Rigshospitalet
Genetic department
Gl landevej 7
Glostrup, Denmark 2600

Dear Dr. Møller,

Thank you for submitting your revised manuscript entitled "Hyperactive mTORC1 affects cellular localization of Smoothed and inhibits GLI2 activation.". We would be happy to publish your paper in Life Science Alliance pending final revisions necessary to meet our formatting guidelines.

- please address the remaining comments from both Reviewers
- please be sure that the authorship listing and order is correct
- please upload all figure files as individual ones, including the supplementary figure files; all figure legends should only appear in the main manuscript file
- please upload your Tables in editable .doc or Excel format
- please add a Category for your manuscript in our system
- please add the Twitter handle of your host institute/organization as well as your own or/and one of the authors in our system
- please incorporate any points from the Conclusion section into the Discussion; we only allow a Discussion section
- the contributions selected for Elsebet Østergaard do not qualify for authorship. Please either update the contributions in our system and the Author Contributions section of the manuscript or let us know if the author needs to be removed (and added eventually to the acknowledgment section)
- since Figure 8 is a Graphical Abstract, please upload it with the file designation "Graphical Abstract", and remove its legend and callout from the manuscript file
- please add your main, supplementary figure, and table legends to the main manuscript text after the references section

LSA now encourages authors to provide a 30-60 second video where the study is briefly explained. We will use these videos on social media to promote the published paper and the presenting author (for examples, see <https://docs.google.com/document/d/1-UWCfbE4pGcDdcgzcmiuJl2XMBJnxKYeqRvLLrLSo8s/edit?usp=sharing>). Corresponding or first-authors are welcome to submit the video. Please submit only one video per manuscript. The video can be emailed to contact@life-science-alliance.org

To upload the final version of your manuscript, please log in to your account:
<https://lsa.msubmit.net/cgi-bin/main.plex>

A. FINAL FILES:

B. MANUSCRIPT ORGANIZATION AND FORMATTING:

****Reviews, decision letters, and point-by-point responses associated with peer-review at Life Science Alliance will be published online, alongside the manuscript. If you do**

want to opt out of having the reviewer reports and your point-by-point responses displayed, please let us know immediately.**

Sincerely,

Reviewer #2 (Comments to the Authors (Required)):

Revised manuscript LSA-2023-02419-TR-A, 'Hyperactive mTORC1 affects cellular localization of Smoothed and inhibits GLI2 activation' by Larsen, Ostergaard and Moller describes an investigation of primary cilium biogenesis in Tsc2 KO mouse embryonal fibroblasts (MEFs). The authors show that Tsc2 KO affects Hh signaling, and that this is likely due to defective signaling through the mechanistic target of rapamycin (mTOR) complex 1 (mTORC1). Specifically, Tsc2 KO MEFs show reduced Gli1 mRNA expression that can be rescued by re-expression of Tsc2 or by down-regulation of mTORC1. The manuscript advances the field by showing that Tsc2 KO, like Tsc1 KO, affects Hh signaling in MEFs. The authors conclude however that KO of Tsc2 and Tsc1 have distinct effects on GLI2 expression and localization and speculate that this might affect the response of individuals with TSC to therapy. The authors present a large number of experiments, and have generated a considerable amount of data. I am sorry, but even with the revisions and clarifications, the manuscript was still very difficult for me to follow. This reflects my lack of knowledge regarding regulation of the primary cilium, particularly in relation to mTORC1 signaling. There were numerous grammatical errors; I assume that the editor will be able to correct these. Furthermore, I had some minor, specific comments that the authors should be able to address:

1. Figure 1. Normalized Gli1 mRNA expression in untreated Tsc2 KO MEFs appears reduced compared to wild-type (WT) cells in Figure 1A and 1B, but not in Figure 1D. Why is this? This was confusing for me. For clarity, I suggest including Figure S1 as part of Figure 1 and to include charts showing the fold induction of Gli1 mRNA expression due to the Shh (Figure 1B) and siRNA (Figure 1D) treatments. If space is an issue, then I would suggest including the immunoblots as supplementary data.
2. Figure 1. On the immunoblots TSC2 is shown as an approximately 140 kDa protein. My understanding is that it should be significantly larger: 180 - 200 kDa. Is there a reason for this discrepancy?
3. Results: Effect on downstream transcription factors (page 8) and Figure 4. It is stated that Gli2 mRNA expression was increased upon purmorphamine (Pur) and rapamycin (Rapa) treatment (Figure 4A). I think this statement is misleading. Pur treatment did not result in a significant increase in Gli2 expression, and combined

treatment with Pur and Rapa did not result in a significant increase in Gli2 expression compared to Rapa only.

4. Results: Effect on downstream transcription factors (page 8) and Figure 4. It is stated that Rapa treatment did not lead to any additional increase in GLI2 protein. Was this also the case for the Tsc2 KO MEFs treated with Pur? I miss the statistical comparison in Figure 4C, and I think that the immunoblot suggests that there is indeed an increase in GLI2 expression upon Pur/Rapa treatment. Did the authors quantify the immunoblot signals?

Reviewer #3 (Comments to the Authors (Required)):

This revision has addressed my major concern regarding the efficiency of ciliation as a confounding factor in observed changes in Hh pathway activity. Putting the data from this manuscript and the data in previous papers published from the same group, it seems that the variation in ciliation observed here is less likely to contribute to the pathway activity.

For the introduction part, it is advisable to rewrite the part on the role of PTCH1. The perspective presented here represents the earlier view on the relation between PTCH1 and SMO, but accumulating data later suggests that ciliary exit of PTCH1 is not required for SMO entry. Therefore, it is better to write this part focusing on PTCH1 activity: In the absence of Hh-ligand, PTCH1 is active inside the primary cilia, posing a constitutive inhibition on SMO. When Hh-ligand binds to PTCH1, PTCH1 activity is blocked, allowing SMO activation and accumulation in the primary cilia.

2nd Authors' Response to Reviewers

09 August 2024

Dear Editor, Eric Sawey, PhD

We are happy to hear that you will publish our paper in Life Science Alliance

-please address the remaining comments from both Reviewers . We have responded to the remaining comments from the reviewers (see below)

-please be sure that the authorship listing and order is correct: It is correct
-please upload all figure files as individual ones, including the supplementary figure files; all figure legends should only appear in the main manuscript file . We have now added all figure legends in the main manuscript file

-please upload your Tables in editable .doc or Excel format. We believe that the tables (Table 1 and Table S1 are in the format)

-please add a Category for your manuscript in our system: Done

-please add the Twitter handle of your host institute/organization as well as your own or/and one of the authors in our system . We have only a X address for the institute hope this is sufficient @DRigshospitalet

-please incorporate any points from the Conclusion section into the Discussion; we only allow a Discussion section . We have now only a discussion section

-the contributions selected for Elsebet Østergaard do not qualify for authorship. Please either update the contributions in our system and the Author Contributions section of the manuscript or let us know if the author needs to be removed (and added eventually to the acknowledgment section) . EØ has also contributed with Conceptualization and design it is now added to the manus.

-since Figure 8 is a Graphical Abstract, please upload it with the file designation "Graphical Abstract", and remove its legend and callout from the manuscript file The graphical abstract is now designated "Graphical Abstract"

-please add your main, supplementary figure, and table legends to the main manuscript text after the references section. They are now added after References

If you are planning a press release on your work, please inform us immediately to allow informing our production team and scheduling a release date. No

LSA now encourages authors to provide a 30-60 second video where the study is briefly explained. We will use these videos on social media to promote the published paper and the presenting author (for examples, see

<https://docs.google.com/document/d/1-2UWCfbE4pGcDdcgzcmiuJI2XMBJnxKYeqRvLLrLSo8s/edit?usp=sharing>). Corresponding or first-authors are welcome to submit the video. Please submit only one video per manuscript. The video can be emailed to contact@life-science-alliance.org We do not know yet

. FINAL FILES:

These items are required for acceptance. -- An editable version of the final text (.DOC or .DOCX) is needed for copyediting (no PDFs).

-- Summary blurb (enter in submission system): A short text summarizing in a single sentence the study (max. 200 characters including spaces). This text is used in conjunction with the titles of papers, hence should be informative and complementary to the title. It should describe the context and significance of the findings for a general readership; it should be written in the present tense and refer to the work in the third person. Author names should not be mentioned. Provided: The first demonstration that mTORC1 inhibits Hedgehog signaling. mTORC1 prevents cilia accumulation of Smo and activation of Gli2. Also, evidence is provided for a 2-step activation process of Gli2.

B. MANUSCRIPT ORGANIZATION AND FORMATTING: Full guidelines are available on our Instructions for Authors page, <https://www.life-sciencealliance.org/authors>

****It is Life Science Alliance policy that if requested, original data images must be made available to the editors. Failure to provide original images**

upon request will result in unavoidable delays in publication. Please ensure that you have access to all original data images prior to final submission.** 3

The license to publish form must be signed before your manuscript can be sent to production. A link to the electronic license to publish form will be available to the corresponding author only. Please take a moment to check your funder requirements.

Sincerely, Eric Sawey, PhD
Executive Editor
Life Science Alliance
<http://www.lsajournal.org>

Reviewer #2 (Comments to the Authors (Required)): Revised manuscript LSA-2023-02419-TR-A, 'Hyperactive mTORC1 affects cellular localization of Smoothed and inhibits GLI2 activation' by Larsen, Ostergaard and Moller describes an investigation of primary cilium biogenesis in Tsc2 KO mouse embryonal fibroblasts (MEFs). The authors show that Tsc2 KO affects Hh signaling, and that this is likely due to defective signaling through the mechanistic target of rapamycin (mTOR) complex 1 (mTORC1). Specifically, Tsc2 KO MEFs show reduced Gli1 mRNA expression that can be rescued by re-expression of Tsc2 or by down-regulation of mTORC1. The manuscript advances the field by showing that Tsc2 KO, like Tsc1 KO, affects Hh signaling in MEFs. The authors conclude however that KO of Tsc2 and Tsc1 have distinct effects on GLI2 expression and localization and speculate that this might affect the response of individuals with TSC to therapy. The authors present a large number of experiments, and have generated a considerable amount of data. I am sorry, but even with the revisions and clarifications, the manuscript was still very difficult for me to follow. This reflects my lack of knowledge regarding regulation of the primary cilium, particularly in relation to mTORC1 signaling. There were numerous grammatical errors; I assume that the editor will be able to correct these. Furthermore, I had some minor, specific comments that the authors should be able to address:

1. Figure 1. Normalized Gli1 mRNA expression in untreated Tsc2 KO MEFs appears reduced compared to wild-type (WT) cells in Figure 1A and 1B, but not in Figure 1D. Why is this? Responds: Yes, it is actually strange, it could be due to some amount of TSC2. We have now included in the paper: No reduction in the basal level of Gli1 mRNA was observed, possibly because

Tsc2 siRNA treatment does not lead to a total loss of TSC2 (Figure 1D, insert).

This was confusing for me. For clarity, I suggest including Figure S1 as part of Figure 1 and to include charts showing the fold induction of Gli1 mRNA expression due to the Shh (Figure 1B) and siRNA (Figure 1D) treatments. If space is an issue, then I would suggest including the immunoblots as supplementary data. Respond: As the focus of the paper is on the total amount of GLI1 mRNA and not the fold induction (which goes in the opposite direction due to the very low basal level of GLI1 in TSC2^{-/-} cells), we have decided not to include these results in Figure 1 but only in figure 1S. We believe that including fold induction in Figure 1 will confuse the reader unnecessary. We hope you agree. We have included in the text: Indeed, as a result of significantly lower basal level of Gli1 mRNA in the Tsc2^{-/-} cells, compared to the WT cells, the fold increase in expression of Gli1 mRNA after Pur and SHh stimulation was significantly higher in the Tsc2^{-/-} cells compared to the WT cells (Figure S1).

2. Figure 1. On the immunoblots TSC2 is shown as an approximately 140 kDa protein. My understanding is that it should be significantly larger: 180 - 200 kDa. Is there a reason for this discrepancy? Respond. Thank you so much. It was a typo now corrected to 200 kDa.

3. Results: Effect on downstream transcription factors (page 8) and Figure 4. It is stated that Gli2 mRNA expression was increased upon purmorphamine (Pur) and rapamycin (Rapa) treatment (Figure 4A). I think this statement is misleading. Pur treatment did not result in a significant increase in Gli2 expression, and combined treatment with Pur and Rapa did not result in a significant increase in Gli2 expression compared to Rapa only. Repond. You are totally right we have now changed it to: Investigation of the expression level of Gli2 mRNA in Tsc2^{-/-} MEFs revealed that it was in fact increased as an effect of Rapa treatment, whereas no significant effect on the GLI2 mRNA level was observed as an effect of Pur treatment 4. Results: Effect on downstream transcription factors (page 8) and Figure 4. It is stated that Rapa treatment did not lead to any additional increase in GLI2 protein. Was this also the case for the Tsc2 KO MEFs treated with Pur? I miss the statistical comparison in Figure 4C, and I think that the immunoblot suggests that there is indeed an increase in GLI2 expression upon Pur/Rapa treatment. Did the authors quantify the immunoblot signals? Yes actually we have performed the experiment 7 times (see legend to figure : Insert: A representative western blotting is shown (n=7, P= 0.085).

Reviewer #3 (Comments to the Authors (Required)): This revision has addressed my major concern regarding the efficiency of ciliation as a confounding factor in observed changes in Hh pathway activity. Putting the data from this manuscript and the data in previous papers published from the same group, it seems that the variation in ciliation observed here is less likely to contribute to the pathway activity.

For the introduction part, it is advisable to rewrite the part on the role of PTCH1. The perspective presented here represents the earlier view on the relation between PTCH1 and SMO, but accumulating data later suggests that ciliary exit of PTCH1 is not required for SMO entry. Therefore, it is better to write this part focusing on PTCH1 activity: In the absence of

Hh-ligand, PTCH1 is active inside the primary cilia, posing a constitutive inhibition on SMO. When Hh-ligand binds to PTCH1, PTCH1 activity is blocked, allowing SMO activation and accumulation in the primary cilia. Respond We how now changed the introduction according to your instruction

RE: Life Science Alliance Manuscript #LSA-2023-02419-TRR

Dr. Lisbeth Birk Møller
Rigshospitalet
Genetic department
Gl landevej 7
Glostrup, Denmark 2600
Denmark

Dear Dr. Møller,

Thank you for submitting your Research Article entitled "mTORC1 hampers Hedgehog signaling in Tsc2 deficient cells". It is a pleasure to let you know that your manuscript is now accepted for publication in Life Science Alliance. Congratulations on this interesting work.

DISTRIBUTION OF MATERIALS:

Again, congratulations on a very nice paper. I hope you found the review process to be constructive and are pleased with how the manuscript was handled editorially. We look forward to future exciting submissions from your lab.

Sincerely,
